# Adult Mesenchymal Stem Cells from Oral Cavity and Surrounding Areas: Types and Biomedical Applications

**DOI:** 10.3390/pharmaceutics15082109

**Published:** 2023-08-09

**Authors:** María Eugenia Cabaña-Muñoz, María Jesús Pelaz Fernández, José María Parmigiani-Cabaña, José María Parmigiani-Izquierdo, José Joaquín Merino

**Affiliations:** 1CIROM—Centro de Rehabilitación Oral Multidisciplinaria, 30001 Murcia, Spain; mecjj@clinicacirom.com (M.E.C.-M.); parmigianijm@gmail.com (J.M.P.-C.); jmparmi@clinicacirom.com (J.M.P.-I.); 2BIONORDIC S.L, 47151 Valladolid, Spain; mpelazfernandez@gmail.com; 3Departamento de Farmacología, Farmacognosia y Botánica, Facultad de Farmacia, Universidad Complutense de Madrid (U.C.M), 28040 Madrid, Spain

**Keywords:** stem cells, dental mesenchymal stem cells (DMSC), cell therapy, dental pulp stem cells (DPSC), biomedical applications of stem cells, Stem Cells Derived from periodontal dental ligament (PDLSC), cerebral ischaemia, cardiovascular diseases, regeneration, dentistry, neuroregeneration

## Abstract

Adult mesenchymal stem cells are those obtained from the conformation of dental structures (DMSC), such as deciduous and permanent teeth and other surrounding tissues. Background: The self-renewal and differentiation capacities of these adult stem cells allow for great clinical potential. Because DMSC are cells of ectomesenchymal origin, they reveal a high capacity for complete regeneration of dental pulp, periodontal tissue, and other biomedical applications; their differentiation into other types of cells promotes repair in muscle tissue, cardiac, pancreatic, nervous, bone, cartilage, skin, and corneal tissues, among others, with a high predictability of success. Therefore, stem and progenitor cells, with their exosomes of dental origin and surrounding areas in the oral cavity due to their plasticity, are considered a fundamental pillar in medicine and regenerative dentistry. Tissue engineering (MSCs, scaffolds, and bioactive molecules) sustains and induces its multipotent and immunomodulatory effects. It is of vital importance to guarantee the safety and efficacy of the procedures designed for patients, and for this purpose, more clinical trials are needed to increase the efficacy of several pathologies. Conclusion: From a bioethical and transcendental anthropological point of view, the human person as a unique being facilitates better clinical and personalized therapy, given the higher prevalence of dental and chronic systemic diseases.

## 1. Introduction

Dental cavities and periodontitis, or periodontal disease (PD), are among the most common diseases of mankind. The epidemiological data are alarming: cavities in permanent teeth are found in over 2.3 billion people, and periodontitis affects over 750 million, and together, they are considered the major cause of tooth loss [1]. The two oral diseases impact health [2]. Dental cavities are lesions in the dental enamel that, if they spread, affect the underlying dentine with an infection that can reach the dental pulp; their development is a consequence of microbial growth supported by sugar and the carbohydrate metabolism induced by a diet that provokes acidification, altering the homeostasis of dental mineralization and damaging the structures through demineralization [3]. Periodontitis is chronic and progressive, characterized by the expansion of the microbial biofilm at the edge of the gums, where it establishes an inflammatory infiltrate that helps destroy the union between the connective tissue and the tooth; reabsorption of alveolar bone; and even causing tooth loss, clearly affecting the individual’s quality of life. Additionally, rising numbers of ever more clinical and scientific studies warn of the correlation between both infectious processes and certain systemic diseases that act as comorbidities, such as cardiovascular diseases, diabetes, rheumatoid arthritis, neurodegenerative diseases, adverse outcomes in pregnancy, cancer, and SARS and CoV-2, through still poorly understood molecular and cellular mechanisms [2,4,5,6,7,8,9]. In fact, the daily clinical practice of a dentist could be summarized very easily and concisely as a battle against dental plaque bacteria and the consequences of polymicrobial biofilm imbalance. The different specialties approach this battle from different angles, whether direct or indirect, emphasizing the importance of understanding the formation, development, composition, effects, and/or alterations of the oral microbiota and microbiome so as to prevent and/or treat the consequences. Consequently, patients come to the clinic not only to treat infections and aesthetics but also because of traumas, congenital anomalies, genetic problems, salivary gland dysfunctions, tissue alterations, leukoplakia, lichen, orofacial cancer, and other systemic pathologies that have oral repercussions. As a response to this demand, for years the classic treatments to repair and cure have employed different new materials that are not necessarily biocompatible, such as BPA (Bisphenol A), found in composites and dental corrective braces, among others [10], or, on the contrary, not compatible with the organism at all, such as the mercury found in older dental amalgams [11,12]. Even more biocompatible materials like implant titanium and the other materials employed in prosthesis may fail, and as a group of replacement materials, they present a more or less limited duration of functionality and benefit [13].

Today, dentists, after the euphoria of the advances of the last decades in replacing lost teeth with different types of prostheses up to their replacement with dental implants, have contributed to an important improvement in their patients’ well-being. However, this result is not sufficient to eradicate the concern generated by peri-implantitis that arises despite the advantages obtained with osseointegration [14]. Decimated by these infectious diseases that may or may not be accompanied by the host’s particularities and his/her habits, results reports indicate that nearly 50% of the patients have periimplantitis and/or mucositis significant enough to endanger the survival and stability of the implant [15]. The progress in improving osseointegration [16] has not canceled a lack of concern and vigilance in clinical practice, and so refined alternative techniques have been introduced, like tools for removing immobile implants that are as efficient and atraumatic as possible [17]. Consequently, despite improvement in biomaterials, techniques, and biotechnologies, there is dissatisfaction with the present advances, leading to the ever-closer qualitative leap of personalized and advanced precision therapies derived from the use of multipotent stromal cells of dental origin, not only for the dental field but also as potential treatments within the field of regenerative medicine [18,19].

Mesenchymal stem cells (MSC) originating in the teeth and surrounding areas show a strong capacity for self-renovation and differentiation into multiple cell lines in comparison with other mesenchymal stem cells [20,21]. Additionally, they show good properties for tissue engineering given their high proliferation rate, strong potential for odontogenic and osteogenic differentiation, together with excellent immunomodulatory properties [22]. Dentists are at the forefront of patient participation in therapies that can potentially save lives using cells derived from their own stem cells taken from their own temporary and permanent teeth. This is without forgetting that pulp and periodontal lesions are still pressing challenges in dental practice. In fact, the cell potential of one’s own tooth is possible at more than one stage of life [23,24]. At the same time, we must point out that most authors emphasize the non- or minimally non-invasive routine nature of the harvesting technique, which constitutes an advantage compared to other techniques that obtain adult mesenchymal stem cells from bone marrow (BM-MSC), adipose tissue, peripheral blood, iPSC (induced pluripotent stem cells), among others, or sources conditioned by time or state, such as the umbilical cord, placenta, membrane, and amniotic liquid [25,26,27].

In general terms, the importance of these multipotent cells is based on the therapeutic applications that can be achieved given their easy harvesting in both children and adults. Therefore, for the clinician and researcher with experience in regenerative medicine and therapy using tissue engineering, a dental biobank is a guarantee for personalized medicine for those diseases or lesions that may affect a person during the different stages of their life [26,27,28]. At present, there are two main strategies for osseous and dental regeneration based on MSC: the rescue or mobilization of endogenous MSC and the application of exogenous MSC in cythotherapy or tissue engineering. However, despite the important evolution and establishment of safe, effective, and simple methods using stem cells to repair and regenerate bone and teeth, it is still a recognized challenge, particularly when considering the adverse effects of a disease microenvironment [29]. It should be pointed out that stem cells can replicate themselves indefinitely and produce both differentiated and undifferentiated offspring. On the contrary, progenitor cells can only achieve a finite number of cell divisions, they do not renew themselves on their own, and the number of cell types they can generate is limited [30].

## 2. Dental Development

Tooth development during intrauterine life begins with the migration of cells from the neural crest toward the maxilla and mandible within/along the three blastoderm leaves/layers (ectoderm, mesoderm, and endoderm) that give rise to the different tissues and organs. During odontogenesis, these dental bands grow and are soon associated with condensations of mesenchymatous cells, forming the so-called dental buds or germs that evolve into a cup and then bell a stage [31]. The enamel organ is composed of cells called ameloblasts, whose function is to secrete dental enamel. Adjacent to the ameloblast layer lie the dental papilla cells, which are a condensation of the neural crest mensenchyma located on the internal concave surface of the enamel organ and are responsible for the secretion and mineralization of the dental dentine. Once these cells are transformed into odontoblasts, they become highly specialized in dentinogenesis (Figure 1 shows germinar layers).

Therefore, odontoblasts and ameloblasts, respectively, secrete the precursors for dentine and enamel. In this manner, the tooth is formed over the course of months. Composed of a soft tissue dental pulp, it is a reservoir for fibroblasts, accompanied by a unique vascular-nervous system that feeds it from the apex (without accessory vascularization, differently from the other surrounding oral tissues), and is responsible for maintaining homeostasis and the future replacement of damaged odontoblasts [31,32]. For example, once the tooth has emerged, the dental pulp cells respond to microbial aggressions such as caries, chemical substances, traumas, etc., triggering the liberation of inflammatory cytokines that can induce the underlying pulp stem cells to differentiate from odontoblasts and so produce repairing dentine [31,33]. However, when inflammatory processes surpass a light inflammatory capacity, the regenerative and reparative processes cease, and pulp necrosis is produced. To continue the dental development, the tooth is surrounded by a structure of mesenchymal cells called a dental sac, and in turn these cells produce some specialized components for the extracellular matrix that give rise to the radicular cement and the periodontal ligament that allows a firm bond with different types of fibers between the tooth and the maxillary or mandibular bone. The root cement communicates with the bony tissue through the apical papilla. Thus, each tooth has a time for eruption and substitution, grouped into deciduous or temporary dentition, constituted by 20 teeth, while the buds of the permanent teeth continue to develop within the bone and, in their time, will provoke the reabsorption of the root of the deciduous tooth shedding, leaving the space that will be occupied by the permanent tooth [31,32]. Being aware of the timing and watching the times for the replacement of the temporary teeth that begin between 5 and 6 and 7 years of age and end at around 12 years of age, as seen in Figure 1. After the wisdom tooth, the last to erupt, the third molar may appear between 15 and 25 years of age as long as the development of the maxilla permits, but most are impacted within the bone (with different degrees of impaction) due to the lack of space in the dental arches. It is advisable that the doctor be familiar not only with the time of dental replacement but also with the reading of a panoramic X-ray (Figure 1 (right part)) shows an X-ray of the mouths of patients (Figure 1, left).

Consequently, being aware of the development, exchange, and dental eruption times during the different stages of growth is very useful when collecting and storing adult mesenchymal stem cells from the dental pulp, given their embryonal ecto-mesenchymatous origin [34]. The cranial neural crest cells are multipotent stem cells that contribute very substantially to the development of vertebrates and give rise to varied cell and tissue types. From there, the destiny of these cells is to assist in the formation of the condensed dental mesenchyma (dental papilla, odontoblasts, dentine matrix, pulp, cement, periodontal ligament, Meckel cartilage chondrocytes, mandible, temporomandibular joint disc, branchial arch nerve ganglia, etc.) [35]. Thus, each tooth is formed by tissues of differing hardness: enamel, dentine, cement, and soft tissues like the dental pulp and the periodontal ligament. Therefore, MSC from adult tissue form part of organ and tissue homeostasis, as they also supervise and provide cells for repair, and, what is more, the dental pulp-derived MSC not only have the potential to differentiate themselves into the expected mesodermic cell lines but also into ectodermic and endodermic cell lines [36]. The great body of evidence shows the growing attention paid by researchers and clinicians to stem cells from teeth and other alveolo-dental complex structures of dentition.

## 3. Research on Tooth Regeneration and Its Structures: Basic Concepts

Due to their differentiation potential, clonogenic DMSCs are perfectly suited to the field of dental tissue engineering and regeneration. So much so that they have been ideal for the regeneration of enamel, dentin, pulp, and support tissues [37,38]. To make possible the regeneration of functional biological tissues, it is necessary to apply the principles of tissue engineering (interdisciplinary field), based on the use of a triade composed of stem cells, scaffolds, and growth factors [39,40] differentiate certain types of cells, thus maintaining the integrity of those tissues in which they reside, such as skin, blood, tissue pulp, and bone, among others [41,42,43]. Odontogenesis is based on the organized reciprocal interaction of the odontogenic epithelium and tissues derived from the neural crest. On the other hand, the scaffolds are the support of the cells on which they are cultivated in vitro, to be later transplanted together with their matrix in vivo. Similarly, scaffolds can serve for drug delivery, to attract body cells to the scaffold site and form new tissues, to structurally support and transport growth factors, DNA, biologically active proteins, and cells, as well as to provide important physical signals to the scaffolds. biological repair and regeneration processes [44,45]. It is important that the scaffolds mimic the natural extracellular matrix of the tissue to be replaced, and thus an optimal design allows for the regeneration of dental tissue to achieve mechanical integrity and functionality that aid cell adhesion and differentiation. As a third pillar, the growth factors that cells release and are taken up by neighboring cell surface receptors that interact in the extracellular matrix were suggested [37].

To address enamel regeneration, it is important to take several parameters into account. Ameloblasts are the specialized enamel-forming epithelial cells, which are lost by apoptosis after development at the time of tooth eruption, and thus mature enamel becomes acellular; therefore, enamel cannot regenerate on its own. In fact, the artificial materials used to restore enamel defects due to caries, trauma, and others lack the same mechanical, physical, and aesthetic properties as the body’s hardest tissue. However, as desirable as the regeneration or fabrication of tooth enamel may seem, de novo enamel tissue engineering and its potential clinical implementation remain a distant and quite daunting task. Consequently, other sources of ameloblastic stem cells have been explored, including cervical loop stem cells, epithelial rest of Malassez (ERM) cells, induced pluripotent stem cells, and keratinocytes [46]. These cells were seeded in various types of matrices, including scaffolds and collagen gel sponges that favor cell union, proliferation, and differentiation, as well as the formation of calcified tissues [47]. Similarly, several signaling molecules have been proposed to be involved in the epithelial-mesenchymal interactions present during odontogenesis. Bone morphogenetic protein (BMP-2) signaling was shown to be crucial for ameloblast differentiation and enamel formation [44]. Regularization of this and other molecules described below could induce the generation of precursors of the ameloblast lineage that resemble odontogenic MSCs in pathways to enamel regeneration. However, the data obtained from the various combinations and attempts, which the authors refer to in their studies, have approached structures and tissues similar to dental germs, teeth, enamel, and ameloblastic cells, among others. Therefore, to date, enamel tissue engineering remains a unique biotechnological challenge [40,46].

On the other hand, for the regeneration of dentin, it must be taken into account that part of the pulp-dentin complex functions as a single unit, for which the main function of the pulp is the synthesis of matrix that is later mineralized and forms the dentin. Primary dentin is derived from this process, which is initially generated and surrounds the entire pulp until the complete formation of the root. Then there is the secondary dentin, which is established throughout life as a physiological function, and finally the tertiary dentin, which is reactionary and repairs aggressions. Therefore, dentin arises from two different populations: those that come from the original postmitotic odontoblasts and those derived from those generated by the pulp, which are DPSCs, respectively [31,37]. Furthermore, a lineage tracing study demonstrated that the new odontoblasts generated during reparative dentinogenesis in molar dentin in mouse models come from perivascular cells, identified by the expression of α-smooth muscle actin (αSMA). In addition, it was shown that the progeny of the αSMA+ population barely participated in the physiological deposition of dentin, with minimal contribution from odontoblasts during primary dentinogenesis [48]. Biomaterials commonly used as scaffolds can be natural polymers (collagen, chitosan, alginate, hyaluronic acid, etc.), synthetic materials (polyglycolic acid, polylactic acid, polylactic polyglycolic acid, etc.), or hybrids, a mixture between the materials, synthetic and natural (alginate-laponite, natural polymers modified with arginine-glycine-aspartic acid, etc.), as well as bioactive ceramics, for which each system presents its performance as well as its limitations in its use [37,49]. On the other hand, the capacity of hydroxide was reported for calcium as an aggregate of mineral trioxide (MTA) and Biodentine, one as direct pulp capping in the regeneration of the dentin-pulp complex, by which both compounds assist in the formation of tertiary dentin [50]. In addition, the application of biological printing combined with MSCs of dental origin through clinical methods of 3D biomanufacturing and dental tissue regeneration is an alternative to current classic dental restorations. The characterization of novel dentin-derived extracellular matrix hybrid cell-laden hydrogel biolinks synthesized from dentin matrix proteins and alginate showed high imprinting capacity and cell survival [37]. It can be stated that tissue engineering applied to dental pulp makes sense with the combination of pulpal MSCs with a scaffold and their subsequent insertion into empty root canals. Thus, cell transplantation in regenerative endodontics is considered the transplantation of exogenous MSCs loaded onto biocompatible scaffolds embedded with biological signaling molecules into root canals. The sources of cells vary according to the existing vital pulp in the root canal. In clinical cases of pulpitis where the inflammation is under control, the inflamed coronal tissue is removed down to the healthy and viable pulp. The remanent in the root canal could serve as a source of endogenous stem cells. Consequently, pulpotomy, commonly applied to deciduous teeth with the intention of preserving vital pulp, can also be performed on immature and mature permanent teeth. After this action, resident PDLSCs, DPSCs, or SHEDs can exert their intrinsic abilities to initiate pulp dentin regeneration under the instruction of growth factors [51] (see Figure 2). Although most of the investigations carried out on SCs-mediated regenerative and reconstructive endodontics used animal models, initial clinical data in humans has been available for a few years. The possibility of regenerating pulp tissue, starting from necrotic pulp endodontics, by means of stem cells and biomaterials allows access to a field that is not so clinically clear and expensive. In fact, meanwhile, more studies are suggested that provide clear guidelines that include the appropriate and preferable properties of biomaterials for use in regenerative endodontics [52]. In general, different experimental studies have been attempted with the ultimate goal of dental regeneration with the help of bioengineering to return from full function to the replacement of a lost or damaged tooth. However, dental regeneration faces many obstacles, such as normal tooth formation, obtaining consistent roots, and the evident absence of normal tooth eruption towards functional occlusion. In another study, autologous stem cells were loaded onto the scaffolds (HA/TCP and Gelfoam) and implanted into the alveolar cavities. Three months later, the root was formed, to which a pin and a porcelain crown were placed. The root was not sufficiently resistant to biomechanics due to the influence of one of the scaffolding compounds (HA), which reduced the quality of the dentin formed, which was different from natural dentin [53]. However, shortly thereafter, Ikeda et al. provided evidence that a bioengineered tooth has the same hardness as a natural tooth, an eruption with normal gene expression, and stable masticatory potential. Previously, the authors successfully developed a method to bioengineer a three-dimensional germ with vessels and nerve fibers integrated into the oral environment. In fact, this study in a murine model achieved full functionality in masticatory performance, biomechanical cooperation with the rest of the oral tissues, and responsiveness through sensory receptors to noxious stimuli, among others [54]. Therefore, it is important to continue the studies on biomaterials in tissue engineering that serve as support for DMSC, scaffolds with or without drug release, and, on the other hand, the conditioned medium and its exosomes according to the needs to be covered, with the aim of advancing the understanding that underlies the regeneration of bone defects and providing guidance for its future clinical applications. It should be noted that MSCs of dental origin are not only applicable in oral diseases but also in systemic diseases as immunoregulators, mainly through the decrease in the production of inflammatory cytokines and the inhibition of bone resorption by cell-cell contact or by the paracrine pathway [55]. In fact, the main regulators of osteogenesis in stem cells are growth factors β (TGF-β) and bone morphogenetic proteins (BMP) by increasing ALP and matrix mineralization. Similarly, other factors such as insulin and vascular growth factor, in combination with BMP2, regulate osteogenesis in stem cells and are, in turn, influenced by chemical, mechanical, and physical microenvironmental signals in the organism, being part of the stem cell niche. In addition, photobiomodulation positively influences osteogenic proliferation, migration, and differentiation, as well as light, which is an environmental stimulant and is important at a systemic level for circadian rhythms and vitamin processing, among other no less important functions. Clinical trials are in Phase I or combined I/II, which demonstrates their early stages of development and is not limited to hard tissue regeneration, but is broad to infectious diseases, orthopedics, etc. [56].

Mesenchymal stem cells (MSC) are adult stem cells found throughout the body that share a fixed set of characteristics described above. In general, all adult MSC populations of dental origin mentioned in this review can collectively be called mesenchymal stem cells of dental origin (DMSC), and, as such, they are named according to their origin. Progenitor and MSC can be isolated from soft tissues, like dental pulp from either permanent or deciduous teeth, the dental follicle, and the apical papilla, from support tissues that may come from the periodontal ligament, the gingiva, as well as from the osseous alveolar tissue extracted during a surgery, for example, or even from beyond the neighboring tissues that surround the dental alveolar process, such as Bichat’s adipose sacks, the maxillary tuberosity, palatine wrinkles, incisor gingival papilla, salivary glands, and other oral structures [21,41,57,58]. They may even come from inflammatory and infectious processes like periapical cysts, cavities in the dental pulp, and gingival hyperplasia, among others [59,60,61]. Both sets of teeth and their stages at this time constitute a research focus that is tireless, not only due to their accessibility but also because of their practicable availability, such as autologous or allogeneic SC [36], that facilitates the regeneration of both soft and hard tissues. Additionally, it should be mentioned how easily they can be expanded both in vitro and in vivo and maintain their viability for long periods of time, even after cryopreservation [62].

The fact that stem cells act differently depending on the surrounding microenvironment contributes even more to the understanding of the complex immunomodulation they mediate [63]. Exogenous administration of MSC and their migration to damaged tissue give them a singular communication with and participation in the inflammatory environment, with the capacity to modulate, suppress, or potentiate different dysfunctional and/or pathological processes [27,64,65]. There are two main techniques to isolate dental MSC. Any of the DMSC types can be isolated by enzyme digestion or placed on growth plates for culture. Next, cells can be classified using a technique to detect specific markers [36]. The multipotent potential of DMSC is similar to that of bone marrow-derived MSC (BMSC), including the capacity to differentiate into cells that have the characteristics of odontoblasts, cementoblasts, osteoblasts, chondrocytes, myocytes, epithelial cells, neural cells, hepatocytes, or adipocytes [41,42]. In general, stem cells are defined by having two main properties: self-renovation and, once they have divided, some daughter cells give rise to cells that maintain the character of stem cells or else have changed into differentiated cells as a second pathway. Thus, DPSC are distinguished by their capacity to differentiate into both osteoblasts and odontoblasts, thereby playing a fundamental role in the maintenance of bone remodulation in the dentin, as mentioned above [66]. Also, they are easy to obtain and less prone to present DNA damage as a result of exposure to external environmental factors like sunlight, which can be a problem with fibroblasts that are more sensitive [67]. It has also been reported that DPSC have a greater regenerative potential than mesenchymal stem cells derived from bone marrow (BM-MSC), which are known as representative MSC. On the other hand, the frequency of colony-forming cells obtained from dental pulp is higher than in bone marrow (22–70 colonies versus 2.4–3.1 colonies per 10^4^ seeded cells) [68,69]. In summary, according to the data published by biomedical researchers on different aspects and behaviors of dental MSC, these progenitor stem cells participate in multiple dental syndromes beginning with endogenous repair, and their lineage can reach regenerative medicine and tissue engineering. The advantages increase if the characteristics of DMSC differentiation can be used and the factors regulating their differentiation are well controlled [70,71]. In fact, it has been reported that MSC of dental origin can modulate immune responses through intense activity at both a cellular and a secretomic level in their interaction with an inflammatory microenvironment. However, it is necessary to better understand the bidirectionality of that regulation and its participation in the tolerance of these particular cells, including a deeper understanding of the epigenetic modifications in DMSC modulation [32,55,56,63].

The advances provide a general description of key findings in the identification and heterogeneity of DMSC and MSC from the surrounding/neighboring areas, including via the exploitation of the adhered gingival tissue around an extracted tooth, the tooth itself, and even diseased tissue, if any (all of which is an easily accessible source for different cell populations). In the same way that bone and soft tissues can be obtained from perforating a bone to place an implant, tissue can be collected in sterile filters during oral procedures and easily cultured, becoming another MSC source among different possibilities. Even systems for bioengineering and the culture of rapidly developing tissue could be useful as important platforms for the recapitulation of molecular and cellular interactions. An example of this is the evidence of the benefits obtained from DMSCs (DMSCs: dental mesenchymal stem cells) due to their neural crest origin in differentiating into ectodermal lineages, which still belong to the future therapeutic possibilities for neurodegenerative diseases and nerve injuries. Indeed, studies are limited to the characterization of oral DMSCs as NCSCs by expression of neural crest markers (NCSCs) instead of testing their functional behavior. Only a few studies have observed glial differentiation enhanced or applicable in in vivo studies [72]. A systematic revision showed the importance of obtaining stability during the DMSC transplantation procedure based on control, quality, and efficacy of the cell populations, which can only be guaranteed by tackling heterogeneity even within the subgroup of stem cells isolated from human teeth that show significant differences in so far as their cell properties, such as proliferation and differentiation, in regenerative medicine [36]. Another report provides evidence that DMSC-seeded collagen and hydroxyapatite scaffolds are more efficient than unseeded scaffolds for osteogenesis and new bone formation [25]. Therefore, it is important to continue the studies on biomaterials in tissue engineering that serve as support for DMSC, scaffolds with or without drug release, the conditioned medium, and its exosomes according to the needs to be covered, with the aim of advancing the understanding that underlies the regeneration of bone defects and provide guidance for its future clinical applications.

Some authors maintain that under in vitro induced hypoxia conditions, DPMSCs not only dampen dendritic cell differentiation from monocytes, but also recruit monocytes with immunosuppressive potential (macrophage M2 and IL-10 increase). In addition, they present proangiogenic properties and greater resistance to lysis that determine NK degranulation [56,73]. It has been shown that DPMSCs could modulate immune tolerance by increasing CD4+, CD25+, and FoxP3+ Tregs. Although they do not constitute an effective treatment for rejection, they can modulate immune tolerance in vivo [74]. A clinical trial has used the combination of MSCs derived from oral fat pads and a plate of mandibular lateral ramus cortical bone tissue and anterior iliac crest bone tissue, which could improve bone regeneration in bone defects derived from the alveolar cleft. Although the data did not reflect a significant difference, it was observed that the bone tissue groups, together with the MSCs derived from the oral fat pad, presented a greater amount of new bone formation with the closure of the defect compared to the bone tissue group without derived MSCs. Of the oral fat pad [18]. Figure 2 shows all kinds of MSCs from the oral cavity described in this review.

## 4. Translational Applications of MSC from the Oral Cavity

Regenerative Medicine (RM) is a branch of RM that implies knowledge of “cellular and molecular biology to design dental therapies that aim to restore, repair, rejuvenate and regenerate dental tissues” [75]. Human MSCs can leave their unspecialized or undifferentiated states and transform into other mesenchymal lineages. Thus, they can regenerate bone, cartilage, and fat and even become endothelial cells, muscle cells, or neurons under physiological and experimental conditions [41,42,76,77]. However, with the aim of protecting the most fragile life against degenerative diseases, a completely pragmatic mentality does not always recognize the patient from the anthropological point of view of the person, but rather, given a supposed possibility of success, some aspects are usually skipped that protect morality and ethics, both by scientists and by political leaders (195 Dignitas Personae 2022).Therefore, science, in its freedom, must be guided by ethics to guarantee progress in the true well-being of the human being [78,79]. In fact, the adult SCs obtained through legal procedures do not present moral objections; for example, the MSCs from the oral cavity and surrounding areas described in this review meet the premises regarding human dignity without posing any type of ethical problem. Therefore, one must proceed with great rigor and prudence, minimizing potential risks for patients, facilitating mutual confrontation between scientists, and providing complete information to the general public [80]. In summary, it is of vital importance to take into account the premise of a bioethical and anthropological approach to the human person as a unique being as a starting point for clinical and personalized therapy using adult stem cells of dental origin [79]. It should be remembered that dental mesenchymal stem and progenitor cells, in terms of gene and protein expression, present unique biological criteria. So much, so that they reveal a superior regenerative potential compared to MSCs derived from other body tissues, together with the advantages of obtaining them by minimally invasive procedures that make them promising resources in regenerative therapies [81,82]. Although evidence suggests that MSCs are present in almost all human tissues, this rationale makes them responsible for tissue repair, growth, wound healing, and cell replacement as a result of physiological or pathological causes, as exposed in the previous sections [83]. It should be noted that one of the main advantages of MSCs over other types of stem cells is the teratogenicity of iPSCs and the absence of the controversy that embryonic SCs sustain from an ethical and moral point of view [56]. Although evidence suggests that MSCs are present in almost all human tissues, this rationale makes them responsible for tissue repair, growth, wound healing, and cell replacement as a result of physiological or pathological causes, as exposed in the previous sections [83]. The presence of MSCs of oral origin and their qualities, added to their immunomodulatory properties, and the absence of ethical and moral conflicts, together with their ease of obtaining and possible use for autologous cell therapies, make them a therapeutic potential throughout the person’s life, as mentioned earlier [84]. The absence of the controversy that embryonic SCs sustain from an ethical and moral point of view [56]. Indeed, the clinical application of iPSCs derived from dental tissues and differentiated into cells similar to different tissues (hepatocytes, neurons, neural crest, MSC, retinal pigment epithelium, dental epithelium, and odontoblasts, among others) is very limited, and iMSCs (induced mesenchymal stem cells) are preferred since they are multipotent and allow osteogenic, chondrogenic, and adipogenic differentiation. However, despite multiple experimental and preclinical studies and clinically concluded trials, its translation to the healthcare setting is still questionable [85,86,87]. Among the obstacles, the heterogeneity of their populations, variability in their quality and quantity, as well as factors related to the donor, discrepancy in isolation protocols, in vitro expansion, and differences in cell administration methods, dose, and place stand out, all of which are affected by the different drugs and chemicals that usually accompany them. In addition, safety concerns are included due to the potential teratogenic or neoplastic risk and the transmission of infectious diseases, among other no less important complications [85,88]. Also, a current problem is that several of the existing protocols do not meet manufacturing standards for transferability to a clinical setting [72]. In any case, a recent systematic study reflects the effectiveness of oral clinical studies with complete results on DMSC-mediated therapies compared to current methods based on patient evidence and concludes with the efficacy of the use of these cells in various pathologies. Furthermore, MSCs achieve their therapeutic effects not only through direct contact between cells, but also through the release of bioactive factors and secretomes [56,89]. Nowadays, the challenge continues to be to understand the magnitude of MSC preparations that explain the variability in tissue sources in donor-related issues (age, disease, gender, lifestyle, etc.), under conditions of ex vivo culture (isolation tools, culture media, population doublings, dissociation reagents, etc.), in storage and delivery strategies (fresh or frozen product, transport buffer, injection routes and procedures), and other aspects to bear in mind [26,88,90,91].

On the other hand, it has recently been reported, with regard to antimicrobial defense properties, that MSCs are presented as a potential tool in regenerative dentistry and immunotherapy [92]. In this sense, two mechanisms that induce this antimicrobial activity are postulated: a direct mechanism by which the MSCs could produce molecules such as AMP, IL-17, and IDO, and another indirect mechanism where the MSCs are associated with the modulation of the phagocytic activity and the production of numerous chemoattractants that can recruit SC in general. Although the antimicrobial activity of MSCs is limited, efforts to develop better strategies continue to improve their efficiency in clinical applications and particularly in dentistry. In addition, the transfection of MSCs with AMP (regulated by chemical products). Bacteria, inflammatory cytokines, and vitamin D3 promote antibacterial effects and could also enhance regenerative processes [92,93]. In addition, the field of precision medicine, which includes prevention and treatment strategies, takes individual variables into account, thanks to the development of large-scale biological databases and computational tools that allow analyzing these data. However, realistically, it is important to achieve, at least in the near future, the regeneration of dental tissues such as pulp and dentin in specialized clinical practice [94]. A clinical trial used the combination of MSCs derived from oral fat pads and a plate of mandibular lateral ramus cortical bone tissue and anterior iliac crest bone tissue, which could improve bone regeneration in bone defects derived from the alveolar cleft. Although the data did not reflect a significant difference, it was observed that the bone tissue groups, together with the MSCs derived from the oral fat pad, presented a greater amount of new bone formation with the closure of the defect compared to the bone tissue group without derived MSCs from the oral fat pad [18]. Finally, the artificial development of organoids from adult cells is used for purposes ranging from basic to translational research. In this way, we study human development, its different diseases, and their possible treatments. A wide variety of organoids have been developed to date, coming from adult stem cells of the corresponding organs, both from humans (iPS) and mainly from murine animals, without the compromise that embryonic stem cells imply.

## 5. Types of Mesenchymal Stem Cells from the Oral Cavity (DMSC in General): Cellular Markers and Clinical Applications

### 5.1. Dental Pulp Derived MSC (DPSC)

The dental pulp-derived stem cell population known as DPSC (Dental Pulp Stem Cells) is obtained from the dental pulp (lax connective tissue found inside the tooth) from impacted third molars (extracted due to a lack of space) that were first isolated in vitro by Gronthos et al. These dental pulp SC were obtained by digestion using type I collagenase and were identified as MSC due to their similar properties to those of medulla bone stromal cells in their immunoreactivity profile as well as their self-renovative capacity and multidirectional differentiation potential [42]. Later, these cells were transplanted in vivo (using immunosuppressed mice as hosts), and the dental pulp derived cells generated functional dentine and dental pulp tissue [41]. Additionally, their characterization revealed that these DPSC were able to differentiate into adipocytes [41], osteoblasts, and endotheliolcytes [77]. These cells seem like fibroblasts under the microscope, and from the beginning, it was considered that DPSC were similar to BMSC. In fact, the pulp is a soft tissue with high vascularity and innervation that can differentiate into cells called odontoblasts, which generate a mineralized tissue that is surrounded by and conforms to teeth differently from bone marrow, in which the function performed by osteoblasts is to generate bone [41,57]. However, it is important to highlight that the differentiation potential of DPSC is more restricted than that of bone marrow in vivo [57]. DPSCs were applied to the treatment of irreversible pulpitis and pulpal necrosis, periodontal disease, and cleft lip and palate. The efficacy of DPMSC transplantation has been confirmed in all clinical trials, including bone regeneration for dental implant placement [81,95]. Additionally, a human DPSC subpopulation with osteogenic potential capable of forming bone-like tissue in vivo was identified and called human osteoblast-derived stem cells (ODHPSC). Microarrays were used to compare the genetic profiles and gene expression of the pulp-derived osteoblasts [66]. After a large number of passes, MSC enter senescence and begin to lose their stem cell qualities, including the capacity to proliferate. Therefore, the number of passes is crucial, and the total number of MSC passes must be specified in future publications [96]. After five passes, the genetic expression differed between the DPSCs with different origins, with the former showing higher neurogenic markers than the SCs isolated from deciduous teeth [97]. Therefore, in order to achieve the therapeutic objective, it is important to select the cell population by identifying its specific markers, which will be similar when they share the same organ or tissue, as do DPSC and SHED. Analysis of the surface markers on DPSC indicates positive markers for CD29, CS44, CD73, CD 90, CK105, CD117, CD146, CD271, CD166, STRO-1, and Stro-3 and negative expression for monocytic and hematopoietic line markers [27,36]. From this point, dental pulp has revealed itself to be a rich and accessible source of MSC, and its biological potential is currently being deeply investigated, not only for its potential for dental repair but also for its possibility of maintaining the vascular and nervous homeostasis of the tooth. DPSC have unique characteristics, such as the potential to differentiate themselves not only into typical mesodermic cell lines like osteogenic, chondrogenic, or adipogenic cell lines, but also into ectodermic and endodermic cell lines [27]. Therefore, from a clinical aspect, there are high hopes in regard to their angiogenic, neurogenic, and odontogenic capacities [36]. So much so that a recent in vitro study has demonstrated that modifications of cell culture conditions can stimulate the stem cell line from the dental pulp toward osteogenesis or into pericyte-like cells. In fact, when DPSC are cultured in an endothelial growth medium, the level of NG2, a pericyte marker, rises significantly, as does the level of smooth muscle actin, indicating a promising capacity to stabilize vessels and promote vascular maturation [98]. Another in vitro study reported greater activity by alkaline phosphatase and calcium deposits in DPSC cultures than in BMSC cultures, revealing a greater osteogenic potential on the part of DPSC, both in vitro and in vivo, allowing them to be considered a potential future source of cells for bone tissue engineering [99]. However, pulp cells can be exposed to different types of damage, which means a slight trauma/aggression may stimulate the odontoblasts to generate reactionary dentine. In the face of a severe lesion that penetrates the pulp and destroys odontoblasts, resident MSC can mobilize to differentiate into odontoblast-like cells that could generate reparatory dentine to protect the exposed pulp and repair the dentine [33,51]. It even seems that trauma can provide a generic and indistinct stimulus to all the DMSC, independently of their origin, function, or location, that would presumably ensure an accelerated repair so that the origin of the MSC-derived odontoblasts that repair the damage becomes irrelevant [100]. Effectively, tracing the genetic line of the odontoblasts from MSC showed they had originated from pericytes and glial cells close to the odontoblasts and pulp cells that suffered the aggression. Furthermore, dental pulp regeneration in cases of pulpal death in immature teeth is the goal of regenerative procedures that are based on tissue engineering principles and consist of stem cells, scaffolds, and growth factors [51]. During animal model investigations, human MSCs are often isolated from healthy pulp tissue, usually obtained from orthodontically extracted teeth, supernumerary teeth, or third molars—wisdom teeth [101,102]. It has also been reported that DPSC from inflamed pulp shows a capacity to regenerate the dentine-pulp complex, although the result is weaker than when this is performed by healthy pulp [45,103]. However, another study revealed that stem cells from exposed pulp are more likely to differentiate in the direction of osteoblastic cells than continue to behave as dentinogenic cells [104]. Mesenchymal stromal stem cells (MSC) are a structural and major connective component found in all cells; they detect microenvironmental signals and act to maintain tissue homeostasis. It seems that different molecular elements and processes will influence in greater and lesser measure odontogenic differentiation [32]. An example is that abusive alcohol (ethanol) consumption can affect the capacity of dental pulp cells to repair and regenerate tissues. It has been demonstrated that ethanol reduces phosphatase alkaline activity. The formation of mineralized nodules and suppresses the expression of the odontoblastic marker alkaline phosphatase and dental sialophosphoprotein (ALP and DSPP, among others), as well as its effort in the differentiated profile [63]. So much so that the donor’s health and age, inflammatory surroundings, and even low oxygen levels can also significantly affect the efficacy of a clinical transplant of these cells. Preconditioning the cells with specific molecules is one of the ways to decrease these limitations [105], and these treatments attempt to improve the microenvironment to restore endogenous SC function and growth [29]. These bioactive compounds are plant extracts or active principles from plants that have been tested in vitro with *Sapindus mukorossi* seed oil and AFA algae in patients [105], which are alternatives for treating vital pulp that increase alkaline phosphatase activity and DPSC mineralized nodule secretion. These compounds improve odontogenic and/or osteogenic differentiation and secretion from the vesicles in the DPSC matrix with odontogenic induction, thus stimulating the formation of reparative dentine. In combination with the implantation of stem cells, they are used extensively in hard tissue engineering [51]. Another study into this type of repair has demonstrated that treatment with a low-power non-ionizing laser can reactivate endogenous growth factors like TGFβ-1 in mineralized tissue to stimulate dental progenitor differentiation into odontoblasts and promote tissue repair [106]. At the same time, when quitosane hydrogel is applied to a blood clot with photobiomodulation therapy (PBMT laser therapy, with a light that acts within the wavelengths of 600–1000 nm), it could improve earlier results on dental pulp regeneration [107]. An in vivo study reported that an ectopic transplant in a rat ischaemia model revealed that the CD31 cells from pulp that had been cultured in a conditioned medium induced higher angiogenesis, neurogenesis, and regeneration compared with the effect of CD31 cells from the bone marrow and adipose tissue [108]. In 2011, Giovindasamy et al. demonstrated for the first time that DPSC could differentiate into a pancreatic cell line and be used as autologous SC therapy in diabetes [109]. On the other hand, since SHED are developed earlier, they are thought to have better differentiation potential than DPSC from adult teeth. However, Majumdar et al. demonstrated that, in a rat Parkinson model after a transplant of SC in striated muscle, the DOSCs had greater neuronal plasticity than dopaminergic neurons that SHED, as represented by the improvement in behavioral disturbances [110]. A recent systematic revision showed the importance of obtaining stability during the DMSC transplantation treatment based on control, quality, and efficacy of the cell populations, which can only be guaranteed by tackling heterogeneity even within the subgroup of stem cells isolated from human teeth that show significant differences in so far as their cell properties, such as proliferation and differentiation, in regenerative medicine. That is why it is basic to manage and combine the known nowadays. DMSC markers and their clinical applications [36]. DPSC have had various clinical applications in regenerative medicine, e.g., as treatments for retinal degeneration, spinal cord lesions, Parkinson’s disease, Alzheimer’s disease, cerebral ischaemia, myocardial infarction, muscular dystrophy, and immune diseases [51,81,111,112,113].

In addition, it seems that lipopolysaccharide (LPS) treatment could significantly alter the biological functions of exosomes derived from DPSCs preconditioned with LPS that promoted Schwann cell proliferation, migration, and odontogenic differentiation [114]. Even aged DPSCs still have active cellular metabolism and secrete functional exosomes that can penetrate the blood-brain barrier, suggesting that they might be an effective drug carrier for the treatment of various diseases, especially neurological disorders, cancer, and pulmonary disease, among other pathologies. Likewise, therapy based on DPSC exosomes is promising for the treatment of systemic diseases. For example, it is believed that they would be more efficient in the treatment of neurodegenerative diseases than MSCs from bone marrow or adipose tissue [115]. In fact, the applications of dental pulp MSCs beyond the head and neck have allowed an important place in translational regenerative medicine [13]. Likewise, DPSCs stand out for their great angiogenic capacity to generate structures similar to capillaries through the secretion of angiogenesis regulatory molecules under certain environmental conditions, which are the most important factors together with their excellent faculty in neural differentiation, which make possible functional regeneration of the dental pulp [116]. However, the mechanisms and molecular interactions underlying the exact odontogenic and systemic processes and differentiation are still awaiting elucidation. In fact, the applications of dental pulp MSCs beyond the head and neck have allowed an important place in translational regenerative medicine [13]. Likewise, DPSCs stand out for their great angiogenic capacity to generate structures similar to capillaries through the secretion of angiogenesis regulatory molecules under certain environmental conditions, being the most important factors together with their excellent faculty in neurodifferentiation, which make possible functional regeneration of the dental pulp [116]. However, the mechanisms and molecular interactions underlying the exact odontogenic and systemic processes and differentiation are still awaiting elucidation. It should be noted that MSCs of dental origin are not only applicable in oral diseases but also in systemic diseases as immunoregulators, mainly through the decrease in the production of inflammatory cytokines and the inhibition of bone resorption by cell-cell contact or by the paracrine pathway [55]. In fact, the main regulators of osteogenesis in stem cells are growth factors β (TGF-β) and bone morphogenetic proteins (BMP) by increasing ALP and matrix mineralization. Similarly, other factors such as insulin and vascular growth factor, in combination with BMP2, regulate osteogenesis in stem cells and, in turn, are influenced by chemical, mechanical, and physical microenvironmental signals in the organism, being part of the stem cell niche. In addition, photobiomodulation positively influences osteogenic proliferation, migration, and differentiation, as well as light, which is an environmental stimulant important at a systemic level for circadian rhythms and vitamin processing, among other no less important functions. Clinical trials are in Phase I or combined I/II, which demonstrates their early stages of development and is not limited to hard tissue regeneration, but is broad to infectious diseases, orthopedics, etc. [56].

#### 5.1.1. DPSC and Reversion of Pulpitis

DPSCs were applied to the treatment of irreversible pulpitis and pulpal necrosis, periodontal disease, and cleft lip and palate. The efficacy of DPMSC transplantation has been confirmed in all clinical trials, including bone regeneration for dental implant placement [81,95]. The current approach to treating inflammation and/or infection of the dental pulp is complete removal of the pulp from the root canal by mechanical debridement, chemical disinfection, and then obturation of the root canals with thermoplastic filling materials. However, Nakashima et al. in 2017 presented the first clinical study that achieved dental pulp regeneration. Its aim was to evaluate the regenerative potential of MSCs derived from pulp tissue (autotransplantation) in five patients diagnosed with irreversible pulpitis. Clinically, the patients underwent endodontic treatment (emptying the root canals) on the teeth that would be subsequently treated with autologous stem cells, previously obtained from the extraction of a third molar (wisdom tooth) obtained with consent. The study was carried out with the execution of an optimal and necessary triad for regenerative endodontics that includes the PDMSCs, associated with the specific growth factor SDF-1 alpha (Stromal-derived Factor 1 from stromal cells), in an ideal framework or scaffold to allow the regeneration sought. After four weeks, a sensitivity response was verified through a pulpometer (electrical tests). At 24 weeks, a positive and robust response of pulp tissue sensitivity was demonstrated, suggesting functional reinnervation of sensory signals, recovery of vascular supply as assessed by MRI and CT scan. Likewise, it was possible to observe the presence of regenerated pulp tissue in the root canal and the formation of functional dentin [117]. As well as the option of hosting the cells by chemotaxis of the host’s own endogenous cells, although residual inflammation is anticipated and the periapical region presents challenges for pulpal regeneration, chemotaxis-induced angiogenesis could provide the native defense mechanism that overcomes residual infection and leads to a vital tooth with regenerated dental pulp [118]. Similarly, other molecules, such as simvastatin (SIM), a drug commonly used to treat hyperlipidemia, were reported to improve odontogenic differentiation and accelerate the formation of mineralized tissue and de novo dentin formation both in vitro and in vivo. The combination of SIM with DPSC in canine Beagle dogs improved coronal pulp regeneration as well as dentin regeneration quickly and efficiently [119]. In relation to the data obtained and the success of different clinical trials, the clinical case of a 50-year-old patient with a diagnosis of irreversible symptomatic pulpitis in an upper third molar was addressed. The clinicians extracted the inflamed dental pulp to be isolated and cultured in the laboratory. Subsequently, they obtained L-PRF (fibrin rich in platelets and leukocytes) from the patient’s blood and inoculated the clot with the expanded DPSCs that were introduced into the previously instrumented and disinfected canal. The upper part of the canal was sealed with Biodentine plus a resin filling material. The authors reported that normal clinical and radiographic vitality tests persist after 3 years and refer to personalized stem cell treatment as a potential alternative procedure for the treatment of pulpitis in mature permanent teeth [23]. Among the signaling molecules, BMP-2 controls odontoblastic differentiation of dental pulp stem cells, and transforming growth factor-β (TGF-β) can stimulate odontoblast-like cell differentiation and mineralization mediated by DPSC [120].

Another clinical trial was carried out with stem cells from the dental pulp of autologous deciduous teeth that were transplanted in patients who had suffered traumatic dental injuries; the authors reported on the innervation, vascularization, and complete three-dimensional regeneration of the pulp tissue with termination of the length of the root and closure of the apical foramen in 26 patients. After the 24-month follow-up, the patients treated with DPSC presented no adverse events, which suggests that regenerative therapy is beneficial to treat dental injuries due to trauma [121]. This review focused on clinical findings in the dental field. We only describe here some concluded clinical trials, although they included trials in the recruitment phase, or even without results yet [94]. In this sense, Song, Wen-Peng et al., 2023, updated clinical information on trials with MSC stem cells from the oral cavity. The irreversible pulpitis was reverted by autologous dental pulp transplantation into the root canal plus gelatin sponge and G-CSF (*n* = 5) since their pulp sensibility increased at 1, 2, 4, 12, 24, 28, and 32 weeks after transplantation. In this NCT03386877 clinical trial, autologous dental pulp cells were implanted into bone defect sites (pulp micrografts plus collagen sponge, *n* = 1), and repair effects were observed 6 and 12 months after transplantation [94]. These results offer a real practical consideration in the absence of more randomized clinical trials with a larger number of patients to make it possible to establish regenerative endodontics based on MSCs of pulpal origin within the routine of dental clinical practice [116]. They regulate dental (epithelium-mesenchymal) interactions as well as cellular activities during parental migration and cell fate decisions [32]. On the other hand, the large-scale forces generated by mastication and the bone tissue that surrounds the teeth, among other conditions, are also necessary for the correct formation of the tooth. In any case, exhaustive, in-depth research is needed on the mechanisms that regulate dental (epithelial-mesenchymal) interactions as well as cellular activities during parental migration and cell fate decisions [32]. The large-scale forces generated by mastication and the bone tissue that surrounds the teeth, among other conditions, are also necessary for the correct formation of the tooth. Finally, it should be noted that single-cell transcriptomics is revealing the heterogeneity of stem cell populations, including multiple cell origins and differentiation potentials [75].

#### 5.1.2. DPSC and Cranio-Maxillofacial Bone Defects

The formation of new bone is related to angiogenesis, and stem cells of dental origin participate directly in angiogenesis by differentiating into endothelial cells, apart from stimulating the formation of blood vessels through paracrine angiogenic factors [122,123]. A clinical trial has been reported suggesting that the combination of MSCs derived from oral fat pads and a plate of mandibular lateral ramus cortical bone tissue and anterior iliac crest bone tissue could improve bone regeneration in bone defects derived from the alveolar cleft. Although the data did not reflect a significant difference, it was observed that the bone tissue groups, together with the MSCs derived from the oral fat pad, presented a greater amount of new bone formation with the closure of the defect compared to the bone tissue group without derived MSCs from the oral fat pad [18]. Given the different maxillofacial surgical needs, the development of new valid biomaterials is required for the purpose of replacement, filling, or regeneration of oral and craniofacial bone structures, both in adults and in the growing patient. In this way, the burden of autologous grafts is dispensed with or alleviated, although not always free of complications. They are commonly used during traditional reconstructive techniques to restore shape, function, and aesthetics. A recent study concluded the clinical importance of developing sheets of DPSC cells that could be used as natural three-dimensional structures in the approach to bone loss, especially in those critical bone defects or any type of bone defect in patients suffering from detrimental diseases such as osteoporosis, diabetes mellitus, or drug-related osteonecrosis of the jaw [124]. The conclusions of this study were derived from the analysis of the potential of DPSC-like SHEDs to maintain their undifferentiated state and their osteogenic capacity when arranged in cell sheets. In fact, different types of scaffolds are being developed and tested every day to facilitate the outstanding osteogenic differentiation capacity that DMSCs exhibit in general. Thus, stem cell-based therapies require cells that can either stimulate endogenous differentiation or differentiate themselves. From this fact derive the various therapeutic applications, both in regenerative medicine and dentistry, in combination with bioengineering [40].

#### 5.1.3. Immunomodulatory Effects and COVID-19

MSCs of dental origin possess potent immunomodulatory functions that make them a potential new immunotherapeutic tool for a variety of autoimmune and inflammatory diseases [41,42,76,125]. Following numerous studies, in vitro and in vivo, the effect of such cells depends on multiple factors, such as the tissue source of the MSCs, the experimental setting, and the type of preparation of the immune cells. In fact, cellular dialogue can not only influence the local immune response, but also extend to the systemic level, influencing the properties of both the innate and acquired immune systems [93]. It is important to keep in mind that DPSC and SHED from patients with systemic pathologies (immunological, diabetes, rheumatoid arthritis, etc.) show decreased bioactivity. On the other hand, it has recently been reported, with regard to antimicrobial defense properties, that MSCs are presented as a potential tool in regenerative dentistry and immunotherapy [92]. The main mechanisms may involve the secretion of soluble factors, such as prostaglandin E2 (PGE2), indolamine 2,3-dioxygenase (IDO), transforming growth factor-β (TGF-β), and human leukocyte antigen G5 (HLA-G5), and interactions between MSCs and immune cells such as T cells, B cells, macrophages, natural killer (NK) cells, and dendritic cells [74,116,126]. It should be noted that the regulation of immunity by MSCs in different pathologies could be ambiguous and influence both the ability to eliminate pathogens and collateral tissue damage [93]. In this sense, two mechanisms that induce this antimicrobial activity are postulated: a direct mechanism by which the MSCs could produce molecules such as AMP, IL-17, and IDO, and another indirect mechanism where the MSCs are associated with the modulation of the phagocytic activity and the production of numerous chemoattractants that can recruit SC. Although the antimicrobial activity of MSCs is limited, efforts to develop better strategies continue to improve their efficiency in clinical applications, particularly in dentistry. In addition, the transfection of MSCs with AMP (regulated by chemical products), bacteria, inflammatory cytokines, and vitamin D3 promotes antibacterial effects and could also enhance regenerative processes [92,93]. In addition, it seems that lipopolysaccharide (LPS) treatment could significantly alter the biological functions of exosomes derived from DPSCs preconditioned with LPS that promoted Schwann cell proliferation, migration, and odontogenic differentiation [115]. In addition, some authors maintain that under in vitro-induced hypoxia conditions, DPMSCs not only dampen dendritic cell differentiation from monocytes, but also recruit monocytes with immunosuppressive potential (macrophage M2 and IL-10 increase). In addition, they present proangiogenic properties and greater resistance to lysis that determine NK degranulation [56,73]. It has been shown that DPMSCs could modulate immune tolerance by increasing CD4+, CD25+, and FoxP3+ Tregs; although they do not constitute an effective treatment for rejection, they can modulate immune tolerance in vivo [74]. However, the specific underlying mechanisms and clinical trial results are unclear, but research continues on appropriate doses and concentrations based on DPSC in order to increase safety and efficacy in order to treat current clinical disorders. On the other hand, some studies speculated that stem cells of dental origin could be established as a possible therapeutic option for SARS-CoV-2 treatment in patients (established as a pandemic in 2019), characterized as a severe acute respiratory syndrome [91]. The usefulness of DPSCs stands out for their immunomodulatory characteristics on the host’s immune cells; thus, they could reduce and prevent inflammation induced by the cytokine cascade or regulate it down, as well as their advantages in the regeneration of lung tissues. damaged [91,127]. Along these lines, a recent clinical trial reported on the safety and efficacy of DPSC from allogeneic donors applied to severe cases of COVID-19 with relevant data on clinical improvement and laboratory tests [91,128].

#### 5.1.4. DPSC and Neurodegeneration

The data support the idea that therapy based on stem cells derived from dental pulp (DPSC) represents a possible option in the treatment of inflammatory and degenerative diseases of the central and peripheral nervous systems [129]. An intrahippocampal injection of human DPSCs was even shown in an in vivo study to generate fully developed blood vessels without the need for scaffolding, containing perfectly aligned endothelial cells (mesoderm), basement membranes, and pericytes, following grafting into the brains of immunocompromised mice [130]. Even aged DPSCs still have active cellular metabolism and secrete functional exosomes that can penetrate the blood-brain barrier, suggesting that they might be an effective drug carrier for the treatment of various diseases, especially neurological disorders, cancer, and pulmonary disease, among other pathologies. Likewise, therapy based on DPSC exosomes is promising for the treatment of systemic diseases. For example, it is believed that they would be more efficient in the treatment of neurodegenerative diseases than MSCs from bone marrow or adipose tissue [115]. In fact, the applications of dental pulp MSCs beyond the head and neck have allowed an important place in translational regenerative medicine [13]. Likewise, DPSCs stand out for their great angiogenic capacity to generate structures similar to capillaries through the secretion of angiogenesis regulatory molecules under certain environmental conditions, which are the most important factors, together with their excellent faculty in neural differentiation, which make possible functional regeneration of the dental pulp [116]. However, the mechanisms and molecular interactions underlying the exact odontogenic and systemic processes and differentiation are still awaiting elucidation. In fact, the applications of dental pulp MSCs beyond the head and neck have allowed an important place in translational regenerative medicine [13]. It was even shown in an in vivo study that an intrahippocampal injection of human DPSCs generated fully developed blood vessels without the need for scaffolding, containing perfectly aligned endothelial cells (mesoderm), basement membrane, and pericytes, following grafting into the brains of immunocompromised mice [130]. Likewise, DPSCs stand out for their great angiogenic capacity to generate structures similar to capillaries through the secretion of angiogenesis regulatory molecules under certain environmental conditions, which are the most important factors, together with their excellent faculty in neurodifferentiation, which make possible functional regeneration of the dental pulp [80,116]. However, the mechanisms and molecular interactions underlying the exact odontogenic and systemic processes and differentiation are still awaiting elucidation. There is also speculation about extracellular vesicles derived from MSCs, loaded or not with drugs, being developed as ready-to-inject biologics [131]. As an example of this, a study investigated the survival of neuronal cells and axonal regeneration through soluble factors secreted by DPSCs (present risk of injury to the trigeminal nerve and its peripheral branches in dental practice due to third-party extraction, impacted molars, implant placement, etc.). Even DPSCs and their extracellular vesicles have the capacity to induce angiogenesis, which is essential in the regeneration of injured central nervous tissue [130]. In addition, Askari et al. observed that oligodendrocyte progenitor cells derived from DPSCs have favorable therapeutic potential for treating sciatic nerve injury in a rodent model [129]. The neurotrophic growth factor secreted by DPSCs can be easily collected, purified, and stored, thus avoiding complications associated with stem cell treatment, such as unwanted proliferation and/or differentiation to unexpected cell types [132,133].

In recent years, many types of stem cells have been used, such as in multiple attempts at therapies for Alzheimer’s disease (AD), an example of which is the advantages in neuroprotection obtained from DPSCs used in in vitro models [134]. DPSCs are known to derive from the ectodermal-neuroepithelial-neural crest lineage, which produces many cell types, including cells of the peripheral neural system and glia. Therefore, continuing with biomedical research, favorable effects are currently obtained due to the safe and effective evidence of MSC-based therapies in animal models that, due to their ease of handling and isolation, have supported the approval of several clinical trials in humans [135]. Similarly, it has been shown that human DPSCs express a neuronal phenotype and produce neurotrophic factors and bone morphogenetic protein 2 (BMP-2), whereby human DPSC treatment significantly increases cell viability and reduces apoptosis in an in vitro AD cell model, with the morphological appearance of neurons restored by elongated dendrites, microfilaments, and dense, well-organized fibrils [136]. On the other hand, the antigenic determinant (part of a macromolecule recognized by the immune system). Tau is the major component of neurofibrillary tangles (NFTs); therefore, antibodies against Tau and NFTs have allowed the characterization of the pathogenesis of neurofibrillary tangles (NFTs) in Alzheimer disease (EA). A study that has emerged for the first time describes an endogenous Tau protein in human DPSCs detected by its phosphorylated and unphosphorylated epitopes, demonstrating its potential to model normal Tau status and the possibility of inducing changes that mimic the modifications associated with neurodegeneration in AD [137]. In any case, despite the promising advances and preclinical studies in transgenic mouse models of AD, this has not been the case with clinical trials in humans due to the multifactorial nature and complex pathophysiology present in AD. In this sense, a multimodal vision with a pharmacological approach as well as the stimulation of neurogenesis and endogenous and exogenous synaptogenesis are jointly postulated for progress in the improvement of people suffering from Alzheimer’s disease. Even small-molecule inhibitors of glycogen synthase kinase 3 (GSK3) have been used for the treatment of neurological disorders in clinical trials such as Alzheimer’s disease by stimulating the formation of reparative dentin with new naturally generated dentin at the damage sites [138].

##### DPSC and Cerebral Ischaemia

In clinical studies using adult stem cells derived from human BS-MSCs, they have been shown to be safe and well tolerated, but without significant clinical improvement in ischemic stroke [139]. On the other hand, stem cells from dental pulp are widely available and accessible and may contribute to well-established therapies for stroke, which would allow for extended therapeutic time and/or impact. DPSCs differ from the rest of the adult stem cell populations due to their embryonic origin from the neural crest and are of special interest due to their neurotropic character, which makes DPSCs and their exosomes especially attractive as a new therapeutic tool for relief of stroke symptoms and other potentially neurodegenerative diseases [140]. It has been reported that intravenous administration of DPSC-type stem cells after reperfusion in a rodent model of cerebral ischaemia is capable of promoting post-infusion functional recovery and improving neurological deficits in cerebral ischaemia. In fact, the evidence postulates that the beneficial effects of transplantation with DPSC in murine models of neurological diseases promote beneficial effects in models of cerebral ischaemia due to occlusion of the middle cerebral artery (MCAO) [141,142]. In general, cell transplantation performed by the intravascular route of administration by MCAO in cerebral ischaemia is less invasive than the intracerebral or intraventricular route [105]. The benefits are based on the reduction of microglial activation as well as the reduction of proinflammatory cytokine levels after reperfusion. Some authors even suggest that DPSCs are more suitable than those of the BM-MSC type [143]. These transplanted DPSCs can be differentiated with high efficiency into neural phenotypes since they express neuronal markers such as β tubulin-III. On the other hand, DMSCs increase the survival of cardiomyocytes under hypoxia in vitro [140]. These studies demonstrated that DPSCs could migrate and survive in the damaged area of the brain, offering suitable therapy for brain repair, either through pre-differentiation and replacement of lost neurons or inducing endogenous neuronal survival by paracrine factors [144]. These studies demonstrated that DPSC could migrate and survive within the lesioned area of the brain, offering an appropriate therapy for brain injury, either through pre-differentiation and replacement of lost neurons or through paracrine-mediated endogenous neuronal survival advocates [144]. In general, cell transplantation by the intravascular route of administration is less invasive than the intracerebral or intraventricular route in rodent models of cerebral ischaemia by MCAO [105]. An in vivo study reported that an ectopic transplant revealed that the CD31 cells from pulp that had been cultured in a conditioned medium induced higher angiogenesis, neurogenesis, and regeneration in a rat ischaemia model compared with the effect of CD31 cells from the bone marrow and adipose tissue [108,145].

Clinical challenges are often accompanied by effects on age, comorbidities, subtype, and stroke severity, which may affect the efficacy and safety of stem cells [142]. The NCT04608838 trial (double-blind, placebo-controlled, multicenter, phase 1/2 clinical trial by multicenter: J-REPAIR) is the first human study that evaluates the efficacy and safety of JTR-161 by allogeneic human dental pulp stem cell administration in patients with acute ischemic stroke in Japan (from January 2019 to July 2021). Patients with a clinical diagnosis of stroke and a National Institutes of Health Stroke Scale (NIHSS) score of 5–20 at baseline were enrolled in this clinical trial. JTR-161 is a novel allogeneic human dental pulp stem cell (DPSC), an isolated product from the extracted teeth of healthy adults). Previously treated patients with recombinant tissue-type plasminogen activator and/or endovascular thrombectomy were allowed to be enrolled. The study consists of three cohorts: cohorts 1 and 2 (each with 8 patients) and cohort 3 (60 patients). Subjects were randomly assigned to receive either JTR-161 or placebo in a 3:1 ratio (1 × 10^8^ cells administered in cohorts 1 and 3 × 10^8^ cells administered in cohort 2), and the cells were administered in a 1:1 ratio in cohort 3. The primary endpoint was the proportion of patients who achieve an excellent outcome as defined by all of the following criteria at day 91 in cohort 3: modified Rankin Scale ≤ 1, NIHSS ≤ 1, and Barthel Index ≥ 95. This therapy product’s safety and efficacy were important when given as a single intravenous administration within 48 h of symptom onset [144]. Nowadays, another trial aims to evaluate a new allogeneic human DPSC product in post-acute ischemic stroke patients, applied as a single dose within the first 48 h of symptom onset [144]. Clinical challenges are often accompanied by effects on age, comorbidities, subtype, and stroke severity, which may affect the efficacy and safety of stem cells. In any case, more studies are required to establish precise and adjusted protocols for patients with the first symptoms of cerebral ischaemia, as judged by neurologists and neurosurgeons involved in real first-line care for cerebral ischaemia. Clinical challenges are often accompanied by effects on age, comorbidity subtype, and stroke severity, which may affect the efficacy and safety of stem cells [142]. The researchers assume that the results of this double-blind study (DPSC) could establish it as a novel therapeutic option [146]. In any case, more studies are required to establish precise and adjusted protocols for patients with the first symptoms of cerebral ischaemia, as judged by neurologists and neurosurgeons involved in real first-line care for ictus and acute myocardial infarction [146,147,148,149].

#### 5.1.5. DPSC and Research in Cardiovascular Diseases

Cardiovascular diseases (CVD) represent one of the leading causes of death in the Western world. The main driver of CVD progression is aging, as it induces vascular changes and can reduce the regenerative potential of stem cells. In turn, CVDs also affect progenitor cells, causing increased senescence and reduced proliferation [147]. This could represent a limitation for autologous stem cell transplantation in CVD; however, it is possible to solve this variable by using allogeneic MSCs, selected based on age and comorbidities [88]. Increased life expectancy carries the risk of developing a wide variety of metabolic and inflammatory disorders that represent a public health problem. Gandía et al. reported that DPSC could be useful for repairing infarcted myocardium [148]. After injection of DPSC into a rat model of acute myocardial infarction, cardiac function improved and infarct size was reduced. In fact, it is probably due to its ability to secrete proangiogenic and antiapoptotic factors. In addition, SHED-conditioned medium has a therapeutic effect on acute cardiac injury by suppressing inflammation and apoptosis [139]. The initiation of allogeneic adult stem cell therapy in age-related chronic diseases clearly warrants further research, although it might provide new therapeutic strategies against CVD and acute myocardial infarction. In clinical studies using adult stem cells derived from human BS-MSCs, they have been shown to be safe and well tolerated, but without significant or important clinical improvement [149]. On the other hand, stem cells of dental pulp origin are widely available and accessible and may contribute to well-established therapies for stroke, which would allow for extended therapeutic time and/or impact. DPSCs differ from other adult stem cell populations due to their embryonic neural crest origin and are of special interest due to their capacity for neurotropic factor release, which makes DPSCs and their exosomes especially attractive as a new therapeutic tool for pain relief from the symptoms of stroke and other potentially neurodegenerative diseases [140]. The researchers assume that the results of this double-blind study on the use of DPSC could establish it as a novel therapeutic option [146].

#### 5.1.6. DPSC and Advances in Diabetes

Type 2 diabetes (DM2) is a chronic metabolic disease that constitutes a risk factor for the development of vascular diseases and causes a high mortality rate worldwide. DM2 has been reported to have adverse effects on stem cell function, inhibiting the angiogenic capacity of MSCs through downregulation of proangiogenic factors. Even BM-MSCs from diabetic patients show impaired paracrine secretion and a greater propensity to differentiate into adipocytes [88]. Several preclinical studies have identified the ability of MSCs derived from the dental pulp of permanent teeth to be deciduous in different in vivo models of pathology, with satisfactory results. However, there are only published data from a few clinical trials carried out to date that have shown beneficial effects in patients with various pathologies, ranging from dental diseases to diabetes to some autoimmune diseases. In 2011, Giovindasamy et al. demonstrated for the first time that DPSC could differentiate into a pancreatic cell line and be used as autologous SC therapy in diabetes [97]. Likewise, advantageous results were obtained on the therapeutic capacity of DPSC and SHED in a murine model induced by streptozotocin. The data showed recovery from normoglycemia in the transplanted mice, whereby it was reported that DPSCs differentiate into pancreatic cell lineages and might be suitable for autologous MSC therapies in diabetes [146]. Similarly, Hata et al., using a single injection, observed the therapeutic effects of transplanting DPSCs from human third molars in a mouse model: improved nerve conduction velocity, decreased blood flow, and increased sensory perception thresholds. Furthermore, hDPSC-conditioned medium induced neurite outgrowth in dorsal root ganglion neurons. Its effect was even prolonged, which could be beneficial in the long-term treatment of diabetic polyneuropathy [150]. In addition, the immunomodulatory effect of DPSCs, which was demonstrated in a diabetic rat model, could reduce diabetic polyneuropathy [27]. In another study, the possible antidiabetic effect of SHED cell administration was evaluated in a mouse model of diabetes. After receiving treatment with DPSC and SHED-type stem cells in diabetic animals, their glucose levels reverted to normal values [151]. In addition, the complications associated with diabetes in a model of diabetic nephropathy were lessened since pancreatic damage is reduced and renal function improves once the DPSC-type cells are administered. Another noteworthy fact associated with DPSC treatment was the formation of pancreatic islets and the increases in insulin levels observed 30 days after the transplant with DPSC. Likewise, transplantation with DPSC improved renal function, as evidenced by the reduction in proteinuria and the normalization of urea levels to normal reference values. It even reduced neuropathic pain in diabetic rats [152,153]. Another study investigated the potential for transdifferentiation of matrigel-grown human PLSCs with agents inducing pancreatic islet cell differentiation to form three-dimensional clusters. Some works validated their differentiation into functional cells capable of secreting insulin in response to high glucose concentrations, similar to pancreatic islets with alternative repair [154]. Finally, optimization of pancreatic organoid culture to maintain pancreatic progenitors and functional β cells, or islets of Langerhans, could revolutionize the treatment of diabetes [155]. Regarding recent studies on stem cell therapies for the treatment of diabetic neuropathy, the results show promising potential for the regeneration of damaged tissues [156].

#### 5.1.7. DPSC for Reducing Liver Disease and Corneal Diseases

Liver cirrhosis is an irreversible fibrotic change in the liver; it can have serious consequences such as impaired liver function, portal hypertension, diffuse degeneration, and even hepatocellular carcinoma. Liver transplantation remains the only treatment option to prevent a more serious clinical course resulting from cirrhosis. Cell-based therapies are the new therapeutic alternatives to whole organ allografts [84]. DPSC and SHED have a great capacity to differentiate into cells similar to hepatocytes. DPSC mesenchymal cells acquire hepatic morphological and functional characteristics when cultured in vitro [41]. When these human MSCs were isolated from the follicle, the pulp, and the dental papilla from the extraction of the third molar of a donor, that is, from a single tooth, it was observed that the MSCs can differentiate in vitro into osteoblasts, adipocytes, chondrocytes, and functional cells similar to hepatocytes [41,76]. In addition, the MSCs from the dental germ of a third molar were transplanted into a rat with induced liver injury, as confirmed by serum bilirubin and albumin levels [43]. Even after hepatogenic induction, all MSCs transdifferentiated into hepatocyte-like cells, which implies the ability to store glycogen [157]. The regenerative competence of DPSCs is confirmed by the fact that after their administration, hepatic dysfunctions are recovered in treated mice with carbon terachloride (CCl4: inducer of hepatic alterations).

On the other hand, corneal blindness such as glaucoma affects millions of people worldwide and is currently the first condition to be treated by grafting from cadaveric tissue. Gene and protein expression studies have shown that DPSCs differentiate effectively into keratinocytes in vitro; a corneal stroma-like tissue construct can be generated by tissue engineering and provide function as a keratinocyte in vivo without causing overt rejection [158]. The dental pulp contains a population of adult stem cells similar to the corneal stroma and is presented as a reservoir for future research. Furthermore, SHEDs express markers similar to corneal limbal stem cells, which, when presented in sheets alone or in combination with the amniotic membrane, allowed regeneration of the corneal epithelium in a rabbit model with total limbal stem cell deficiency. Other studies have been described previously in this chapter in the section on MSCs derived from oral epithelium [144]. Ng et al., in a previous study, induced human PDLSCs into the retinal lineage and subsequently modified the induction protocol to target these cells toward retinal ganglion-like lines and determine miRNA signals in the transdifferentiation process. The data revealed that DPSCs exhibited characteristics of functional neurons, formed synapses, and displayed glutamate-induced calcium responses as well as spontaneous electrical activities [159]. Autologous stem cells present a future perspective in the area of personalized regenerative medicine and an alternative to cadaveric tissue grafts. The dental pulp contains a population of adult stem cells similar to the corneal stroma and is presented as a reservoir for future research.

### 5.2. Deciduous Tooth-Derived MSC (SHED)

The population of stem cells from deciduous teeth called Stem Cells from Human Exfoliated Deciduous Teeth, with the English Language acronym SHED, can be extracted from the 20 temporary or milk teeth that are naturally replaced during growth and can be obtained non-invasively or with minimum risk. In 2003, Miura and his collaborators identified SHED cells as clonogenic cells that were highly proliferative and able to differentiate into a variety of cell types, including neural cells, adipocytes, and odontoblasts [76]. Both SHED and dental sack-derived MSC (DPSC) are dentally-originated MSC subpopulations that act as the main odontoblasty source during the formation of terciary dentine after a postnatal lesion, given the incapacity of the pre-existing odontoblasts to produce reparative dentine [60]. SHED show antigens similar to MS and are positive for CD29, CD44, CD63, CD71, CD73, CD90, CD105, CD117, CD146, and CD166, and also embronary SC markers like OCT3/4, NANOG, SSEA-3, SSEA-4, TRA-1-60, and TRA-1-81, and progenitor cell markers CD13 and CD31 [36]. Previously, in vivo studies had shown that SHED cells were not only able to generate dentine but could also induce the formation of bone and survive in a mouse brain with neural cell marker expression [76,160]. Therefore, knowledge of the molecular mechanisms and signaling pathways will allow a more exact selection of MSC subpopulations and the use of exogenous factors to develop this potential. In fact, a study reported that CD105 (ENG), a predictive biomarker for osteogenic potential in both SHED and adipose tissue-derived MSC, could be used to select and enrich these subpopulations with a greater osteogenic in vitro potential, as well as the possibility of using other more complex molecular systems (hsa-miR-1287) to regulate its/their expression and exploit the potential of CD105 that is inversely correlated with the osteogenic capacity of the SHED. If one compares DPSC with SHED, the latter shows higher proliferation rates, osteoinductive capacity in vivo, and the capacity to form spheric groups [76].

#### 5.2.1. SHED and Progress in the Regeneration of Cranio-Maxillofacial Bone Defects

Given the different craniomaxillofacial surgical needs, the development of new valid biomaterials is required for the purpose of replacement, filling, or regeneration of oral and craniofacial bone structures, both in adults and in the growing patient. In this way, the burden of autologous grafts is dispensed with or alleviated, although not always free of complications. They are commonly used during traditional reconstructive techniques to restore shape, function, and aesthetics. A study in rats evaluated the efficacy of human SHEDs in the reconstruction of cranial bone defects, which had previously been characterized in vitro as mesenchymal cells with their cell markers showing osteogenic, adipogenic, and myogenic differentiation. Subsequently, human SHEDs were embedded within a collagen membrane as a scaffold and evaluated together with another collagen-only group without SHEDs. After one month, both groups showed signs of bone formation; however, SHED-treated rats showed more mature bone regeneration as well as the absence of signs of graft rejection, suggesting that SHEDs constitute another potential source of MSCs to correct cranial defects [161]. Similarly, another report provides evidence that DMSC-seeded collagen and hydroxyapatite scaffolds are more efficient than unseeded scaffolds for osteogenesis and new bone formation [25]. Tanikawa et al. recently used SHED associated with a hydroxyapatite and collagen sponge for the first time to treat alveolar defects in patients with cleft lip and palate, achieving satisfactory bone healing results. It is known that when treating these children with cleft lip and/or palate, they require a second and more invasive surgical act (obtaining the iliac crest bone graft, among other types of bone) that carries the risk of presenting eventual consequences such as a greater recovery time, pain, higher surgical and hospital costs, etc. These data are relevant since these MSCs of dental origin show similar properties (proangiogenic) and differentiation capacities as bone marrow (adipogenic, myogenic, neurogenic, and odontogenic potential). In addition, from a single temporary tooth, it is possible to obtain 1 × 10 ^40^ SHED, and after five passages, its number is reduced to only 1 × 10 ^20^ SHED, considering that this number of cells was achieved approximately one month after the extraction of the deciduous tooth. The authors highlight the good interaction and adhesion of 1 × 10 ^60^ cells to the average of two to three biomaterials embedded in them [25]. Once this point is reached, these results could offer stable clinical practice consideration since MSC therapy allows its potential application to be extended to the reconstruction of other injuries in the craniofacial surgical field. Even the data obtained are relevant since they suggest the absence of ectopic bone growth related to immature bone; in fact, it seems to have a similar behavior to iliac crest bone grafting, and even after five years of follow-up of clinical parameters, no complications or dangers associated with it were reported for this new modality of cell regenerative therapy [25]. Song et al., have collected all published clinical data with autologous deciduous pulp implanted into injured teeth or allogeneic deciduous pulp phases I and I in patients. Repair effects have been demonstrated by transplanting stem cells into the root canal, SHED, and other kinds of mesenchymal stem cells from the oral cavity in combination with different biomaterials (gelatin sponge, Hydroxyapatitecollagen) or scadfolds (PEG-PLGA) [94].

#### 5.2.2. SHED for Preventing Neurodegeneration

The same as SHED transplantation, this suggests that these cells have a greater mineralization capacity than DPSC [162], while at the same time their genic expression shows a higher level of embryonary markers [97]. What is more, SHED and DPSC show a higher proliferation rate than BMSC [111]. MSC from deciduous teeth induce multilineage differentiation that includes dopaminergic neurons, odontoblasts, osteoblasts, chondrocytes, adipocytes, hepatic cells, cutaneous cells, and endothelial cells. Additionally, Gunawardena and coworkers have demonstrated that dental-derived stem-cell conditioned media promote hair growth stimulation [163]. Therefore, with regard to the SHED properties being derived from the neural crest, they can differentiate into dopaminergic neurons in vitro. Parkinson’s disease (PD) is a neurodegenerative disorder characterized by the loss of dopaminergic neurons in the substantia nigra, which consequently induces a series of motor and non-motor disorders. It has been possible to efficiently induce dopaminergic neurons from SHED-type cells, which, when transplanted into parkinsonian rats treated with 6-hydroxydopamine (6 OH-DA), show beneficial effects. Given that SHED stem cells, when cultured in vitro with trophic factors such as EGF and bFGF, express proneural genes (Ngn2 and Mash1) or dopaminergic neuron-like markers (dSHED) by BDNF (brain-derived neurotrophic factor), treatment could promote beneficial effects in Parkinson’s. In this regard, it has been shown that DPSC or SHED stem cells can differentiate into dopaminergic neurons and that grafted SHEDs survive in the striatum of parkinsonian rats and reverse neurological deficits. However, the differentiation status of the SHED to be implanted determines the clinical efficiency of the transplant since dopamine levels are higher than those of undifferentiated SHED-grafted rats [164]. Thus, Nguyen et al. investigated the pathological association between dopaminergic neuron (DN) development and mitochondria in three children on the Autistic Spectrum (TEA), using SHED as a specific cellular model for the disease or for the patient. TEA is a disorder with heterogeneous neurodevelopment that is characterized by poor social interactivity, restricted interests, and repetitive stereotypic behaviors. The authors reported the alteration and lack of growth in neurites associated with a decrease in the potential of the mitochondrial membrane and of ATP levels, as well as an alteration in the branching of the DN, suggesting a deterioration in the signaling pathway for BDNF that would imply a possible decrease in the production of intracellular dopamine in these children. It is therefore considered an advantage to collect SHED from the age of 6 that would become naturally available and would provide valuable neurobiological information through the in vitro analysis of ND and infer early strategies for the treatment of the first signs of TEA [165]. On the other hand, the neuroprotective effects of SHED are due to the release of soluble factors (cytokines, chemokines, and neurotrophic factors such as BDNF). Furthermore, another study demonstrated that conditioned medium from SHEDs protects against 6-OHDA-induced toxicity in dopaminergic neurons [166]. A recent study demonstrated that SHED exosomes could change the polarization of the microglia and reduce neuroinflammation derived from cerebral trauma, and its secondary consequences, as well as other neurological disorders. In fact, after co-culturing SHED exosomes with microglia for 48 h in a rat model, the researchers found that the concentration of nitrite and the IL-6 and TNF-α inflammatory factors was lower than in the microglia that were activated alone, indicating that the recuperation of motor function was improved and the cortical lesion was decreased. These observations suggest that MSC-derived exosomes can be an important pillar in the repair of tissue damage and in decreasing the secretion of inflammatory factors by microglia [89]. If we compare the plastic capacity of both types of stem cells (DPSC and SHED), it is known that DPSCs possess superior neuronal plasticity towards dopaminergic neurons compared to SHED in cell cultures [110]. However, SHED cells implanted in the striatum of parkinsonian rats ameliorate behavioral deficits in these rats. Another study shows that neurotrophic factors released by dental pulp cells protect against the neurotoxicity of MPP+ neurotoxin or rotenone (inductors of parkinsonism in rodent models) in dopaminergic neurons [167]. Even in vitro neuronal organoids, a quasi-physiological 3D model, would facilitate precise studies of biological processes, niche functions, responses to drugs, mutations, etc., in Parkinson’s disease [155]. On the other hand, several studies have described differences in the generation of iPSC (induced pluripotent stem cells) from cells with different maturities and placements in tissues. In fact, SHED cells have been demonstrated to be more easily reprogrammed because they are more immature than cells isolated from DPSC. Complete reprograming to a state similar to embrionary cells could be avoided in the function of the proposed use, which would reduce the risks typically found with iPSC [168]. At present, SHEDs are promising stem cells for clinical applications such as dental regeneration, osseous regeneration, pediatric surgical diseases, hepatic insufficiency, neural regeneration, and therapeutic revascularization [169]. Finally, SHED-derived exosomes can be classified into three categories: osteogenesis promotion, neurotrophic properties, and anti-inflammatory effects. As an example of neurotrophic property, exosomes isolated from SHEDs grown on laminin-coated alginate three-dimensional microcarriers protected dopaminergic neurons against 6-hydroxy-dopamine-induced apoptosis, whereas exosomes from SHEDs grown under standard culture conditions had no such effects [170]. Chen et al. evaluated the effect of SHED-conditioned medium (SHED-CM) on the treatment of symptoms and neurological deficits achieved in the experimental model of PD induced by rotenone. The data revealed that the administration of a single intravenous dose of SHED-CM (cell-free) was sufficient to ameliorate neuroinflammation, eliminate α-synuclein (its accumulation in axons and presynaptic terminals induces neurodegeneration in PD), recover mitochondrial damage, and improve motor deficits [164]. Furthermore, other studies speculate that the release of the contents of these extracellular vesicles from hPCy-MSC (considered biological waste) and previously differentiated dopaminergic neurons could represent an intelligent model for in vitro research on PD, which would allow access to new biomarkers, drugs, and application routes [171]. Collectively, this evidence supports the regenerative role of stem cells of dental origin; however, further clinical trials are required to corroborate that they can promote repair effects in Parkinson’s disease. On the other hand, since SHED are developed earlier, they are thought to have better differentiation potential than DPSC from adult teeth. However, Majumdar et al. demonstrated that, in a rat Parkinson model after a transplant of SC in striated muscle, the DPSCs had greater neuronal plasticity than dopaminergic neurons that SHED, as represented by the improvement in behavioral disturbances [110].

#### 5.2.3. SHED Immunomodulation and Its Liver Fibrosis Treatment

It is important to keep in mind that DPSC and SHED from patients with systemic pathologies (immunological, diabetes, rheumatoid arthritis, etc.) show decreased bioactivity [27]. Previously, BM-MSCs were used for various human diseases such as bone fractures, severe aplastic anemia, acute GVHD, systemic lupus erythematosus (SLE), etc. Indeed, immunomodulatory functions were found in DPSCs and SHEDs similar to previously known BM-MSCs. The immunomodulatory advantages of SHEDs compared to BM-MSCs have been reported, for which they present significant effects on the inhibition of Th cells and IL17 in vitro, even highlighting that transplantation of SHEDs effectively reverses SLE in mice, increasing the proportion of regulatory T cells, reducing IL17 levels, and preventing autoimmunity and inflammation compared to BM-MSCs [76,82]. On the other hand, donor SHED cells survive in liver tissues damaged by CCl4 and differentiate into hepatocytes expressing genes specific for human hepatocytes that secrete human albumin, urea, etc. Hepatic differentiated DPSCs suppress liver fibrosis and restore serum levels of alanine transaminase, aspartate transaminase, and ammonia, suggesting a therapeutic use of DPSCs to treat fibrosis. Unfortunately, experimental and preclinical studies remain very limited [172].

### 5.3. Apical Papilla-Derived MSC (SCAP)

The stem cells from the apical Papilla, known as SCAP, are isolated from apical papilla tissue and can be easily separated from an extracted tooth with still developing roots, which is usually available given the common extractions of third molars still in the process of rhizogenesis [94]. The apical papilla is a soft tissue that adheres loosely to the apices of immature permanent teeth that have not completely formed all of their roots and can be easily removed with tweezers/forceps [31,94,173]. Likewise, hybrid hydrogels showed the ability to integrate with acid-soluble dentin molecules, improving SCAP differentiation and effectively modifying the pulp-dentin complex [120].

SCAP can be found in the apical papilla (in the area of the radicular foramen) and represents another unique dental stem cell involved in dental development [173]. SCAP are positive for CD24, CD29, CD34, CD45, CD73, CD90, CD105, CD106, CD146, CD166, and STRO-1 [36], with the potential of becoming cell types derived from three germinal layers. SCAP are involved in pulp healing and regeneration in immature teeth with periodontitis or periodontal abscesses. In a dental self-transplant, SCAP seems to survive due to the minimal vascularization found in the apical papilla [174]. They show greater proliferative potential than DPSC when measured using bromodeoxyuridine (Brdu) and seem to be a source of primary odontoblasts [53]. It was also discovered that this cell population expresses high survival and telomerase levels, molecules that regulate cellular proliferation. In fact, Sonoyama et al., instead of trying to form a complete tooth, managed to generate a biological root with its surrounding periodontal tissue, starting from SCAPs combined with PDLSCs in a mini-pig model [94]. Histologically, Sonoyama et al. described an apical papilla with fewer blood vessels and cell components than in dental pulp and, lying between them, an area that was cell-rich; the surface MSC marker, STRO-1, was found in the apical cells, demonstrating the presence of SCs in the tissue [53]. However, the dental papilla is considered an odontoblast precursor during dental development, so when these differentiated cells deposit primary dentine, the papilla is imbued with the dentinary structure that evolved into the pulpar tissue; the remaining adjacent undifferentiated mesenchymal cells later continue differentiating into odontoblasts that form the radicular dentine. However, these authors clearly distinguish between the source of these odontoblasts and their function in dentine: the apical papilla gives rise to primary and secondary dentine-forming odontablasts, while pulp-derived replacement odontoblasts are responsible for tertiary or reparative dentine [173]. Before, replacement odontoblasts were thought to be odontoblast-like cells, and consequently, as stated earlier, evidence has demonstrated that the replacement odontoblasts derived from the mesenchymal cells underlying the cell-rich zone in the perivascular and perineural sheath areas where MSC are found. SCAP neurogenic potential could arise from the fact that SCAP derives from neural crest cells [41]. Like them, this cell population expresses high survival and telomerase levels, two important molecules in mediating cell proliferation [173]. A recent study suggests that SCAT cells are an optimal dental SC population for peripheral nerve repair. The authors compared the capacity of PDLSC, DPSC, and SCAP in response to glial induction in vitro and their direct support of sciatic nerve regeneration in vivo using PCR and ELISA analyses [175]. The three populations show a capacity to produce different levels of neurotrophic and angiogenic factors that promote nerve regeneration.

Nevertheless, the results highlighted increased genic expression of neurotrophic factors, like BDNF and GDNF by SCAP, with respect to PDLSC and DPSC, and also more BDNF liberation by SCAP and DPSC than by PDLSC. In conditioned media, SCAP cells developed into neurite-producing cells with notable general growth, particularly noticeable in length in vitro. What is more, the SCAP gave optimal results in regenerating 10 mm of rat sciatic nerve in comparison to results with PDLSC and DPSC. It is supposed that the main contribution of the investigated cell populations in rat sciatic nerve lesions was due more to their secretome than to direct glial differentiation by the cells. In fact, the cells distinctly grouped in front of the proximal regeneration, and BDNF was detected in the vicinity of transplanted human stem cells, although there was no positive staining for the S-100 glial marker [175,176]. However, it seems that CD24 is a specific SCAP marker that is not detectable in other mesenchymal stem cells, including DPSC and BMSC. These data support the suggestion that SCAP are a unique postnatal stem cell population [53]. These cells can be isolated and cultured efficiently through enzymatic digestion (collagenase type I) or by explant culture, in which case papillary tissue is cut into 1 mm^3^ samples and seeded on culture plates [177]. Until now, SCAP showed the expression characteristics of MSC markers: self-renovation, proliferation, migration, differentiation, and immunosuppression, allowing SCAP to participate in regenerative medicine, even immunotherapy, and dental, bony, cartilaginous, neural, hepatic, and vascular tissue regeneration [178]. SCAP-derived exosomes stand out for their great potential for oral tissue regeneration. Zhuang et al. demonstrated that these exosomes promoted dentinogenesis of BM-MSCs both in vitro and in vivo, indicating that they could represent a potential therapeutic tool for the regeneration of the dentin-pulp complex [179]. In addition, SCAP-derived exosomes injected (palatal gingival area) into a critical size defect mouse model significantly enhanced angiogenesis and soft tissue regeneration [180].

### 5.4. Dental Follicle Derived MSC (DFSC)

Dental follicle cells are a group of MSC that form slack connective tissue, forming a sack that surrounds the dental papilla and the enamel organ, that is to say, it surrounds the tooth bud/seed during the first stages of life (see Figure 1). As a tissue derived from the ectomesenchyme that originated in the cranial neural crest, it is directly responsible for the cementation, the periodontal ligament, and the formation of the alveolar bone in the process of tooth development, as well as being the central regulator of the process of tooth sprouting with the resorption and formation of the alveolar bone [31,181]. As has been demonstrated, cells isolated from the follicular sacs of human third molars are characterized by CD13, CD29, CD44, CD49d, CD56, CD59, CD90, CD105, CD106, CD166, and STRO-1 [36]. These DFSC express high CD9, CD10, CD13, CD29, CD44, CD49d, CD56, CD59, CD90, CD105, CD106, CD166, and STRO-1 levels [36]. DFSC show a greater proliferation capacity, and their protein profiles are mostly similar to those of neural crest cells. In this manner, they are able to differentiate into osteoblasts and osteoclasts in the temporal genic space. At the same time, the development of the root requires epithelial stimulation of Hertwig’s epithelial sheath (HERS), a bi-layered structure that develops from the internal epithelium and the external enamel. This epithelial-mesenchymatous interaction induces DFSC differentiation into cementoblasts [31,182]. Also, Lima et al. demonstrated that dental follicles contain a significant proportion of neural progenitor cells that express β-III-tubuline (90%) and nestine (70%). Curiously, an immunocytochemical study demonstrated that DFSC were positive for the neural crest markers p75b (50%), HNK (<10%), and a small proportion (20%) of glial cell markers (GFAP), being the first study to report the presence of neural crest SC (NCSC) and glial-like cells in the dental follicle [183]. On the other hand, a comparative in vivo study unexpectedly found that the regenerative capacity of the periodontal ligament was greater in DFSC than in PDLSC. The DFSC layers, after being transplanted to fill periodontal defects, showed a greater regenerative capacity for cementing and periodontal union, although the authors report that further investigation is necessary to elucidate the regulatory mechanisms [184]. From the perspective of in vitro cell studies, most are performed in static conditions, although one study noted the influence of the centrifugal force on the DFSC and DPSC populations loaded on porous 3D scaffolds. The results showed that not only did the innovative dynamic conditions promote more DFSC proliferation, but also increased the expression of osteogenic activity-related genes while decreasing phosphatase alkaline activity and inducing a greater deposition of osteopontin in comparison with static culture conditions [185]. As has been shown, the different cellular culture conditions should be carefully chosen since their effects can affect the later clinical application. DFSC, as well as bone, cement, and periodontal ligament, show a capacity to regenerate tissue that is similar to pulp and dentine, which makes it possible for them to regenerate a tooth root. In fact, a study combined DFSC with biological scaffolds to transplant them to the alveolar fossa, the omentan sacs, and the cranial fossa. Although mineralized matrix was observed in the omental sacs and cranial fossa, root-like tissue was only formed in the alveolar fossa. This result suggests that the alveolar fossa microenvironment is most adequate for the construction of roots when DFSC is applied [186]. Later, in computer-assisted design (CAD), a root complex was constructed combining DFSC on a scaffold with a treated dentine matrix; this radicular biological complex not only successfully regenerated a root-like structure, but it also withstood the masticatory function and remained stable after being restored with a crown in the following three months [187]. Also, according to the data reported by Park et al., it is possible to cryopreserve the dental follicle tissue as a long-term DFSC resource. For this purpose, follicles were subjected to development with a specific cryoprotectant and a slow ramp speed freezing process [188]. These characteristics increase the potential applications of DFSC in tissue engineering, particularly in the orofacial region, including the repair of the alveolar bone, periodontum regeneration, and the formation of bio-radicular complexes [189]. Likewise, various osteogenesis methods have been tested with the purpose of improving osteogenic stimulation and differentiation of MSC odontoblasts derived from dental tissues. Indeed, an original study aimed to stimulate DFSC, DPSC, and APSC to differentiate into osteoblasts using CBD (cannabidiol) and compare the results with Vit. D3 (a standard power-up). The data obtained showed that the cellular responses were different to the same stimuli, for which the best osteogenic expression of CBD came from the APSC, unlike the DPSC, which showed better potential and mineralization when treated with Vit. D3. The DFSC and the APSC, in terms of mineralization, presented a good response to low doses of CBD and Vit. D3. Therefore, the immediate clinical applications of CBD could expand its use in osteoinduction in combination with the analgesic effect that characterizes it, making it an adjuvant for bone fractures induced by metastasis and/or osteoporosis. CBD could even increase the survival rate of stem cells afterward in the recipient due to its anti-tumor, antioxidant, and anti-inflammatory properties [190].

### 5.5. Dental Germ Progenitor-Derived MSC (TGPC)

Tooth Germ Progenitor cells were first identified by Ikeda et al., in 2008 in the germ of the third molar in its late bell stage [43]. They reported the characterization and distinctive character of the TGPC and revealed their high proliferation activity and capacity to differentiate in vitro cells from the three germinal layers, including osteoblasts, neural cells, and hepatocytes. Flow cytometry analysis showed that TGPC were positive for CD29, CD44, CD73, CD90, CD105, and CD166 [36]. TGPC were examined using transplantation into a murine model with a hepatic lesion (tetrachloride-treated) to determine if this new cell source could be useful in treating hepatic diseases. The authors suggested that the satisfactory results given its multipotent capacity on the TGPC graft make them good candidates for cell therapy for treating hepatic diseases and offer unprecedented opportunities for the development of treatment for tissue repair and regeneration [43]. There are experimental data that justify the use of dental stem cells as a promising source of primary cells, principally for the engineering of dental tissue. The interaction of MSC with endothelial and epithelial cells is absolutely necessary for intact dental morphogenesis. Dogan et al. cultured human TGSC-derived endothelial-like cells that successfully differentiated into several types of cells with a wide range of functional capacities in vitro under this focus. These findings could offer a practical consideration as a potential SC source for tissue engineering (epithelial and endothelial tissue) and cell therapy. TGPC can differentiate into muscle, cartilage, adipose, nerve, bone, and dental tissue, so they are considered an important alternative source in regenerative medicine [191]. The STRO-1 cell marker is positively expressed in TGPC with notable osteogenic differentiation capacity. They can expand and sustain nearly 60 population duplications, during which they conserve their spindle-shaped morphology and their high proliferation rate. They express STRO-1 and CD markers associated with MSC and show a tendency toward genic expression associated with pluripotentiality (Nanog, Oct4, Sox2, Klf4, and c-myc), indicative of a mesenchymatous phenotype [84,192]. TGPC shows capacity for multilineage differentiation like that of other dental MSCs, including the capacity to differentiate into adipocytes, osteoblasts, odontoblasts, chondrocytes, and neurons. In hydroxyapatite implants (HA), TGPC showed new bone formation in the presence of recently formed boney matrix and a covering of active osteoblasts in a cuboid shape on the surface of the matrix. In vitro, TGPC can differentiate into cells with the morphological, phenotypical, and functional characteristics of hepatocytes, suggesting that TGPC can be used to treat hepatic diseases [192].

### 5.6. Stem Cells Derived from Periodontal Dental Ligament (PDLSC)

Stem cells derived from periodontal ligament, PDLSC, have been known since Seo et al. studied SC from third molar periodontal ligament that were STRO-1 and CD146+ and found that the PDLSC had the capacity to differentiate into different lineages, including cementoblast and adipocyte-like cells [125,193]. The periodontal ligament (PL) is a connective soft tissue that is found as fibers in the cementum of the dental root and the alveolar bone; it connects them and is continuously remodeled to maintain dental functionality. PL provides cushioning and tooth support functions and is a source of dental nutrition, homeostasis, and periodontal tissue regeneration. PDLSC morphology is like that of fibroblasts, capable of forming colonies with high proliferation rates, and as they are able to regenerate cement, collagen fibers, and Sharpey fibers (periodontal ligament fibers that insert into the periosteum of the alveolar bone), it has even been speculated that they could have a tissue regeneration capacity [194]. A comparative study between gum SC (GMSC) and PDLCS showed their potential to differentiate into osteogenic, adipogenic, and chondrogenic lineages, with the capacity to generate new bone after an ectopic transplant. However, human PDLSC had a much more efficient differentiation potential than GMSC, independent of the culture conditions [195]. The PDLSC in STRO-1+ and CD146+ stand out for their capacity to differentiate into cartilage, making them a promising alternative population for cartilage regeneration [36]. Additionally, they are able to differentiate into neurogenic, cardiomyogenic, chondrogenic, and osteogenic lineages [193,196]. Other studies have confirmed the sufficiency of PDLSC to participate in paracrine or directly in nerve lesion regeneration, and in fact, they are able to differentiate toward the three germinal layers when exposed to specific culture conditions. Additionally, they show strong/high positivity to CD9, CD10, CD13, CD29, CD34, CD38, CD44, CD49d, CD59, CD73, CD90, CD105, CD106, CD146, CD166, and STRO-1 [36]. Cell proliferation levels have been reported to increase in the PL after a lesion and during orthodontic treatment, while, on the contrary, resting rates are low in adults and decrease with age, suggesting that PDLSC, like DPSC, are mobilized by specific stimuli. [197]. From another perspective, Zhang et al. developed a tetra-PEG hydrogel-encapsulated sustained-release aspirin (ASA) system as an appropriate microenvironment to support the viability and proliferation of PDLSCs in vitro. The in vivo study demonstrated that both tetra-PEG hydrogel alone and PEG-ASA were capable of promoting PDLSC-mediated bone regeneration in a mouse cranial defect model; however, the PEG-ASA result presented greater efficiency in bone response [198].

Physiologically, the niche occupied by PDLS contains a variety of tissue components, cell populations, and soluble factors that closely regulate the behavior of these MS. The STRO-1+ and CD146+ PDLSC populations seem to be unique populations in human beings, and they do not exist in mice [36]. A recent study has demonstrated that PDLSC exerts immunomodulator effects by suppressing T cell proliferation in a coculture of periodontal SC with human monocyte-derived cell dendritic (CD) exposed to LPS (isolated from *Porphiromonas gingivalis*) [199]. These Gram-negative bacteria induce proinflammatory cytokines like interleukin 1β (IL-1β), IL-6, and IL-8, which induce periodontal tissue destruction and osteoblastic differentiation, forming a periodontal bag that deepens and increases the anaerobic microorganism ecological niche that leads to an immune inflammatory tissue lesion. The authors concluded that STRO-1+ and CD146+ from DPSC significantly decreased CD1b levels in the major non-classical histocompatibility complex in the CD, resulting in defectuous T cell proliferation, demonstrating the promising potential of PDLSC [199]. In pathological conditions like periodontitis, osteoporosis, and others, it can affect differentiation and MSC viability, which evolves into a worsening of the disease and deficient tissue healing. In fact, the latest research suggests that in the periodontal space, MSC are able to differentiate into progenitor cells for cementum, fibroblasts, and osseous cells, so CD146+ PDLSC show improved efficiency in colony formation and osteogenic potential than CD146− cells. PDLSC can be obtained from extracted teeth either through explanted cultures or enzymatic digestion, and their characteristics seem to depend on the extraction place where they are obtained; in fact, cells isolated from the alveolar bone surface show a greater alveolar bone regeneration capacity than cells obtained from the root surface. Through a minimally invasive process like the removal of premolars in orthodontic treatments, scraping the root surfaces, and then preparing them, one can access the periodontal tissue progenitor cells and expand them in vitro, so they constitute an important SC reservoir of cells with clonogenic capacity [86]. Previously, reconstruction of damaged periodontal systems was not possible, and only conventional therapies were able to maintain the remaining tissue. Consequently, the science of tissue engineering in this field has as its primordial objective the restoration of the lost cells and tissues, and for this it is necessary to isolate PDLSC from periodontal tissues [86,200]. PDLSC combined with acetylsalicylic acid favors osteogenic differentiation in/during bone regeneration [198]. In the same manner, photobiomodulation (PBM) improved the proliferation and osteogenic differentiation in PDLSC-inflamed cells; there was no significant increase in terms of cell proliferation. In contrast, there was a significant increase in the expression levels of osteogenic genes as well as alkaline phosphatase activity in the PBM-treated group compared to the controls. PBM is currently considered an acceptable method to stimulate stem cells without non-invasive absorption by photoreceptors that induce the cell response [201,202]. On the other hand, maintaining PDLSC in undifferentiated forms is more advantageous [125]. They even maintain their tissue regenerative capacity, so after recovering, the cryopreservation of the extracted and stored teeth, they are a useful source for future therapeutic techniques [62]. Also, the microenvironments of donors and receptors in cytotherapy and tissue engineering are critical to determining the regenerative efficacy of the transplanted MSC, which is also influenced by the microenvironment during osseous regeneration, the tooth, and cell interactions [29]. On the other hand, the activity of PDLSC exosomes is beneficial for the maintenance of periodontal homeostasis by promoting proliferation, angiogenesis, and osteogenesis, as well as regulating inflammatory responses. The main regulatory functions of PDLSC-derived exosomes include angiogenesis, anti-inflammation, and osteogenesis. In fact, exosome secretion was higher in PDLSCs under inflammatory conditions. In addition, these exosomes derived from inflamed PDLSCs promoted angiogenesis of human umbilical vein endothelial cells by upregulating the expression of the specific vascular marker CD31 and vascular endothelial growth factor (VEGF) [203].

### 5.7. Alveolar Bone-Derived MSC (ABMSC)

Mesenchymal stem cells derived from alveolar bone, known as ABMSC (Alveolar Bone Mesenchymal Stem Cell), were isolated and expanded from human mandibular alveolar bone by Matsubara and colleagues with a success rate of 70% [204]. Achieving complete and functional osseous regeneration in treating osseous defects, fractures, osteoporosis, osteonecrosis, etc., is a challenge for orthopaedic and craniofacial surgeons. Up to now, many studies have been developed with regard to SC isolated from osseous tissue [205]. However, there are important differences between ABMSC and SC obtained from other sites like the iliac crest (BMSC), the latter being the most often required form. In fact, craniofacial bones, including the alveolar bone, are derived from ectoderm (neural crest cells), while other bones, like the bone from the iliac crest, originate from embryonal mesodermic cells [35]. In fact, BM-MSC are outstanding due to the greater chondrogenic and adipogenic potential in iliac bone cells than in those from alveolar bone; however, both show potent osteogenic activity in vitro and in vivo [204]. ABMSC show positive CD29, CD44, CD73, CD90, CD105, CD146, and STR0-1 positive expression [36]. ABMSC can be quite useful in regenerative medicine because obtaining them from the medulla/bone marrow is less invasive, traumatic, or painful than the procedure to harvest the iliac crest bone. What is more, ABMSC can be easily obtained when third molars are extracted or implants are placed since the alveolar bone is usually exposed and accessible during these surgical procedures. Curiously, an in vivo study compared osteogenic differentiation and characteristics for repairing bone defects in ABMSC and BM-MSC as a scaffold focus for bone regeneration in rabbit skulls; they observed cell markers indicative of higher osteogenic differentiation and greater mineral deposits in the ABMSC than in the BMSC [205]. Another study characterized the immunomodulator properties of BMSC isolated from human iliac crest bone marrow in comparison to ABMSC isolated from alveolar bone. On investigating the effects of the BMSC on immune cell function, including those of proliferation, differentiation, and activation (THP-1 monocytes, macrophages, peripherical blood macrophages, and mononuclear cells), the authors highlighted that among the 42 pro- and anti-inflammatory cytokines and growth factors they tested, only IL-6 and MCP-1 were secreted by ABMSC and BMSC at detectable levels. In general, ABMSC secreted less IL-6 and MCP-1 than did BMSC, but the difference was not statistically significant. It seems that the values show greater immunomodulatory properties in ABMSC than in BMSC, with the latter having potent effects on immune cells. They are therefore able to inhibit the activation and proliferation of monocytes and T cells after cultivation with different immune cell types, including THP-1 monocytes, macrophages, and peripheral blood cells [206]. In fact, infusion of these MSC has been involved in clinical trials to treat GVD, graft vs. host disease, rheumatoid arthritis, multiple sclerosis, etc., with immunomodulatory cell therapies. At any rate, debate on their efficacy in this field continues, so larger studies are needed to gain a deeper understanding of the mechanisms involved. On the other hand, an association between the patient’s age and ABMSC proliferation has been observed, determining a decrease in proliferation with the increased age of the donor [204].

### 5.8. Periosteum-Derived MSC (PSC)

Stem cells derived from the periosteum are called Periosteal Stem Cells (PSC). Fell, who described the osteogenic potential of periosteum, successfully cultured the cells and suggested their capacity to form a mineralized matrix [207]. The periosteum is a highly vascularized, specialized connective tissue made up of an external layer and an osteogenic, or internal, area made up of MSC, fibroblasts, osteoblasts, vessels, and nerves. It has been routinely used for some time as an autologous graft in orthopedic surgery (fractures, grafts, transplantation, cell suspensions, etc.). However, its use for dentistry has been more recent; the internal layer of the periosteum has been considered an excellent source for oral cavity stem cells. However, its full potential is still unknown, but it is quite valued and used in guided tissue regeneration and soft tissue healing techniques [208]. Periosteal contexture changes with age, being thicker in children and thinner and less active in adults, although still conserving its capacity to differentiate into fibroblasts, osteoblasts, chondrocytes, adipocytes, and skeletal myocytes [209]. What is more, the periosteum fulfills the principle of tissue engineering requirements as a cell source, a scaffold to hold and deliver cells, as well as a source of growth factors. SC derived from the elderly seem to be comparable to those derived from young individuals, which could be related to their stability after 24 duplications, telomere length, and telomerase activity [210]. These cells have the capacity to express typical MSC markers and to differentiate into adipocytes, osteoblasts, and chondrocytes. Additionally, clonal populations derived from individual adult human periostal cells are reported to have a multipotential mesenchymal property since they can also differentiate, in vivo and in vitro, into skeletal myocyte lineages [211]. The 2009 Colnot study stands out for having demonstrated differences between SC derived from periosteum, endostium, and bone marrow. The periosteum and endostium contribute to osteogenesis, but only the periosteum also contributes to chondrogenesis, making it the only source for chondrocytes [212]. Additionally, PSC applied to the alveolar border and to increase/raise/reinforce the floor of the maxillary sinus showed more reliable positive results in dental implants, with better bone remodeling and less postoperative waiting time, even as an ideal alternative for large bone defects [211]. This could explain why cells derived from the periosteum could be used in tissue engineering, particularly for bone regeneration [21]. Consequently, the osteogenic potential of BMSC, alveolar bone cells, and periostic cells were compared in the context of bone engineering, and it was found that due to their osteogenic capacity, the periostic cells are the most effective in forming new bone, while the BMSC made quantitatively and qualitatively significantly less bone [209,211]. On this basis, the Duchamp de Lageneste et al. study of a rat model of skull defect identified PSC with clonal multipotency, self-renovation, and presence at the vertex of the differentiation hierarchy of bone cells. The authors reported the transcriptional profile by which PSC express different expression signals than other mature BM-MSC and MSC. While BM-MSC form bone from cartilage using the endochondral pathway, PSC do so through the direct intramembranous pathway, giving them a divergent cellular base between the intramembranous and endochondral development pathways; thus, the osseous system presents multiple types of SC, and each group has different physiological functions. However, although there is a clear distinction in intramembranous competency in the PSC, a lesion may induce plasticity or interconversion between the different cell types, so the PSC contributes to the endochondral fracture repair process. These results increase the possibility that PSC will become attractive targets for pharmacological cell therapy to treat skeletal diseases [213]. As a source of oral bone, periosteum as well as alveolar bone can be used to obtain stem cells to form bone, and they can easily be obtained surgically by a general dentist [209]. Another study highlighted the reduced capacity of graft BMSC to form cartilage and bone during skeletal regeneration when compared to that of PSC, although they are derived from common mesenchymal progenitors during development and bone growth, and the importance of pereostine in the process [213]. Understanding the molecular mechanisms associated with the capacity for intrinsic cell repair and cytoskeletal reorganization under mechanical stress through mechanosensitive signaling emphasizes periostine expression, which assures proper collagen fibrinogenesis, matrix organization and promotes osteochondral regeneration [214]. Despite the high regenerative capacity of the osseous system, more research into its origin, recruitment, SC function during osseous repair, anabolic pathways, cell mechanics, periosteal microenvironment, PSC differentiation, etc. is necessary to create new approaches and strategies for bone defects, disease, as well as the physiologic exchange of osteoporosis and post-traumatic care.

### 5.9. Oral Epithelium Derived MSC (OESC)

MSCs derived from oral tissue epithelial cells are an important source of stem cells. Nakamura and colleagues, despite their success in treating ocular surface reconstruction, decided to improve the technique given the risk of allogenic graft rejection, insufficient donor epithelial cells, or low efficacy in cases with bilateral ocular disorders. The experimental study in rabbits concluded that autotransplant using oral epithelial cells cultured from an oral mucosa biopsy was a possible method to reconstruct the ocular surface [215]. Later, the same author reported the results of the first transplantation of autologous oral epithelial cells cultured on an amniotic huma membrane in coculture with fibroblasts; the culture of the oral epithelial layer was transplanted to the damaged ocular surface, and in 48 h, the entire surface of the cornea was repaired and free of the disorder it had presented before the transplants [215]. It was also discovered that the expression of factors related to angiogenesis is regulated in human corneas after cultured epithelial oral mucosa transplantation, suggesting that the expressions of FGF-2, VEGF, PEDF, endostatin, and IL-1 are similar to those found in normal corneas, conjunctive tissue, and corneas after transplantation [101]. Also, a long-term study has demonstrated the results of this technique (transplantation of cultured epithelial oral mucosa) and suggested that it is a very successful treatment for severe disorders of the ocular surface. In fact, a study by Hyun and colleagues demonstrated that culture starting with human epithelial mucosa cells without fibrin was viable as a cell therapy for treating diseases with limbal epithelial cell deficiencies [216].

### 5.10. Gingival Stem Cells after Wounds (GMSC)

Stem cells derived from the gum or gingival tissue, GMSC, were isolated and characterized by Zhang et al. for the first time in 2009 [217]. The human gum is a unique, masticatory, keratinized mucous tissue that is a fundamental part of the periodontal environment and has an outstanding ability to heal and regenerate after wounds. Gingival stem cells (fibroblasts) have the clear advantage that they are easily isolated from the gum tissue adhered to the piece after an extraction or simple biopsy. Gingival fibroblasts are a heterogeneous cell population that plays a crucial role in wound healing. Additionally, gum is a common tissue that is quickly available from all patients and can be sampled repeatedly if necessary. The efficacy of bone regeneration has been evaluated using predifferentiated GMSCs together with a hydrogel scaffold in order to repair the critically sized maxillary alveolar bone defect. In this study, bone regeneration took place between 4 and 8 weeks post-implantation, as compared to control rats [95]. Likewise, MSCs derived from autologous gingiva were seeded on a xenogeneic collagen matrix with the aim of obtaining sufficient tissue in the root covering of dental gingival recessions in a canine model. The data showed a positive trend for the combination with MSC used in the flap technique to cover the defect compared to the technique without stem cells [38]. What is more, gum MSC can be obtained from different oral sites such as the maxillary tuberosity [218], transverse palatal pleats/folds (palatal wrinkles), including the incisive papilla, related to the neural crest; they are positive for nestine within the Meissner corpuscles, which were first described in adult rats by Widera et al. These SC are very plastic and may have the capacity to differentiate into functional neurons and glial cells, which is important in therapeutic studies [219]. Also, GMSC can be obtained from the oral mucous membrane lamina propria; they express mesenchymal stromal cell markers that are positive for neuronal cells and the neural crest, demonstrating and identifying their sheltering of a population of primitive SC with a phenotype like the primitive distinctive neural crest that can give rise to ectoderm, mesoderm, and endoderm lineages [220]. GMSC can even originate in a hyperplastic gum due to the overactivation of MSC in the gums, suggesting they could be an ideal source for immunoregulatory therapies given their easy availability and accessibility [221]. It has been speculated that gum-derived MSC have distinct properties from the other dental MSC given their constant exposure to a unique microenvironment (microflora, masticatory mechanical stimulation, among others) that may have modulated these cells to make them acquire capacity to support and resist the inflammatory environment and its infections [222]. The authors found high levels of MSC-associated markers like CD73 and CD90 in GMSC from inflammatory tissue, suggesting their function in maintaining progenitor properties. The results proved that GMSC can be isolated from healthy as well as sick gingival tissue and that the inflammatory environment could have a clonogenic stimulation effect. Additionally, the GMSC from inflammatory tissue showed reduced osteogenesis and improved adipogenic potential compared to GMSC [223]. In vitro cell proliferation is much faster for GMSC than BMSC, thus ensuring the number of cells necessary for clinical cell therapy [224]. GMSC express positive markers, including CD13, CD44, CD73, CD90, CD105, CD146, CD166, CD271, and STRO-1 [36]. GMSC are not tumorigenic, an additional advantage in their clinical use. They respond differently to growth factors and produce proteins that are specific to the extracellular matrix during the healing process [225]. Progenitor cells and a multipotent MSC subpopulation have been isolated and characterized among gingival fibroblasts [137,144,218]. These fibroblasts have been used to culture induced pluripotent stem cells (IPS) [226]. They characteristically maintain their phenotype stably during culture, as well as normal karyotype phenotype activity and long-term telomerase, demonstrating good clinical safety [227].

The study by Huang et al. demonstrated that CD39 and CD73 could be useful markers to evaluate the GMSC therapeutic effect in autoimmune diseases. Another interesting result of the study is the finding that GMSC are better than BMSC at controlling xeno-GVHD in vivo. The authors concluded that the manipulation of autologous GMSC is an important strategy in therapy for human autoimmune diseases, graft vs. host rejection (a potentially mortal complication after the transplantation of adult SC), organ transplant, and other disorders of the immune system [224]. On the other hand, it has been reported that gingival stem cells could directly inhibit B-cell activation, proliferation, and differentiation and reduce histopathological damage in lupus nephritis [228]. On the other hand, the injection of PDLMSC in an animal model of induced autoimmune encephalomyelitis demonstrated a decrease in inflammation and demyelination in the spinal cord with increased production of neurotrophic factors and reduced release of proinflammatory mediators [229]. In comparison with PDLSC, GMSC are less susceptible to replicative senescence, the deterioration induced by proinflammatory cytokines in the osteogenic potentials in vitro, and the formation of ectopic bone in vivo [195]. In so far as multipotency is concerned, GMSC not only possess the potential of trilineage mesodermic differentiation (osteocytes, adipocytes, and chondrocytes) but also the potential to transdifferentiate themselves into ectodermic and endodermic cell lines, apart from keratinocytes and endothelial and odontogenic cells [217]. Therefore, given their origin in the neural crest, GMSC can be programmed due to their genomic stability in neural progenitor stem cells or cells like the neural crest to induce the regeneration of the facial nerve, sciatic nerve, and spinal cord, as performed in different murine studies [230]. It should be noted that GMSC morphology is stable, and the morphological characteristics are not lost in even high numbers of passages, maintaining their normal karyotype and telomerase activity in long-range cultures without becoming tumorigenic. In fact, these advantages surpass those of BMSC, making them clinically functional and competent in the short term and appropriate for cell therapy in regenerative medicine and tissue engineering [230,231]. Therefore, GMSC could be an alternative source, given their easy access when compared to progenitor cells from the periodontal ligament, which show an innate capacity to differentiate into the different tissues that make up the dental support structures. [223,231]. What is more, GMSC can perform immunomodulatory and anti-inflammatory functions in the in vivo immune system, making it a promising alternative for cellular treatments in experimental inflammatory diseases [217]. The lingual affectation derived from oral cancer (ablative surgery, radiotherapy, or chemotherapy) seriously affects the quality of life of patients. Zhang et al., by combining exosomes from GMSCs with extracellular submucosal matrix from the small intestine, promoted the regeneration and recovery of tongue papillae in a rat model with a tongue defect [198]. Equally complex, sciatic nerve lesion repair methods are often unsatisfactory; on the contrary, the combination of GMSC exosomes with biodegradable chitin channels was injected into a rat model with positive effects. The results demonstrated that the exosomes, obtained from the cell supernatant, improved the proliferation of Schwann cells, the growth of the neuronal axon of the dorsal root ganglion, and the formation of nerve fibers and myelin, which contributed to restoring motor skills, nerve conduction function, and muscle movement [232]. Also, in another study, the behavior of GMSC exosomes in a high-lipid microenvironment suppressed lipid accumulation and transformed proinflammatory macrophages into an anti-inflammatory phenotype by reducing the secretion and expression of inflammatory factors, including IL-6, IL-1β, TNF-α, and a cluster of CD86 differentiation [233].

### 5.11. Carious Dental Pulp Stem Cells (CDPSC)

Stem cells from carious dental pulp, known as CDPSC, were first isolated and cultured in 2012 [234]. Currently, the number of studies on MSC from healthy dental pulp has increased markedly, studying not only their capacity to regenerate dental tissue and odontoblasts but also their differentiation of other earlier described cell lines [41]. Reparative dentine (tertiary or reactionary) is produced by odontoblast or odontoblast-like cell activity that protects the pulp from external aggressions like caries, attrition, or abrasion [45]. Many studies have been developed and examined this possibility, although Ma et al. studied what happened with carious pulp as compared to healthy pulp, which has not been studied before. The researchers reported that SC are present in unhealthy pulp and in fact present a greater proliferative capacity (over the three passes they studied) and higher alkaline phosphatase activity, mineralization, and expression of genes related to osteogenesis and dentinogenesis than DPSC [234]. Neither the molecular response of CDPSC (SC from pulp with deep caries) nor that of DPSC, nor their different biological responses, are yet understood. In any case, a later study by these authors delved into defining the molecular characteristics that differentiate healthy and carious dental pulp SC, comparing the proteomic profiles combined with dimensionality, which separated the different proteins in both samples after their selection and staining [235]. It must be highlighted that this genetic approach systematically allows advances in the quantitative and qualitative understanding of the entire proteome of the SC from different niches and, thus, their self-renovation capacity, differentiation potential, and capacity for regeneration based on the information obtained from the mass spectrometry results. After their study, their data indicated that the differences between DPSC and CDPSC in the expression of proteins were principally involved in the regulation of cell proliferation, differentiation, the cell cytoskeleton, and motility. They suggested that CDPSC have higher antioxidant protein expression, which may protect CDPSC from oxidative stress [235]. On the other hand, another study reported the similarity of results on a comparison between SHED, SC from healthy deciduous teeth, and SCCD, from teeth with caries, with regard to their proliferation rate, surface markers, capacity for differentiation into the expected cell lines (osteogenic, chondrogenic, and adipogenic), and immunophenotypic characteristics. The interpretation of these results would suppose being able to avoid discarding cavitied deciduous teeth with caries since their CDPSC are not compromised and in fact constitute a possible viable SC source that presents clinical relevance in the application of tissue regeneration therapies [236]. This knowledge would allow a more exact evaluation of the effects of the local microenvironment of caries on DPSC and take their different destinies (application or research) into account. It would even seem that CDPSC shows greater osteogenic differentiation and proliferation potential than SC from healthy dental pulp [234].

### 5.12. MSC Derived from Periapical Cysts (hPCy-MSC)

Mesenchymal stem cells from human periapical cysts are known as hPCy-MSC. An unusual alternative for regenerative medicine was the presence of MSC, which dental clinicians considered a biological waste product consisting of the inflammatory and infectious discharge from periapical cysts of MSC. An infrequent alternative in regenerative medicine was the presence of MSC within a dental clinic setting as a biological discharge from the products of inflammation and infection by periapical cysts. Marrelli et al. first isolated mesenchymal stem cells in 2013 from a human periapical cyst (hPcy-MSC). They were described as a subdivision of MSC with an origin in the periapical dental cyst. As stem cells that are capable of self-renovation and potentially differentiate into multiple cell lines, like osteoblasts, adipocytes, and a chondrogenic line [237]. Periapical cysts are lesions that are found in the radicular apex of the affected tooth (inadequate endodontics, anatomical alterations, etc.) that lead to an inflammatory fibrous process (macrophages, lymphocytes, and neutrophils) and granulation tissue that, with time, becomes a cyst [45,238]. Recently isolated hPCy-MSC express markers like those of other dental MSC, i.e., the main neuronal markers like tubulina β-III and the astrocyte markers like GFAP (glial fiber acid protein). The CD146 marker that regulates osteogenic differentiation is notable [239]. The low CD146 population had a significantly greater osteogenic differentiation capacity than the same CD146 if it was high, which means that CD146 plays an important role in the regulation of the properties of hPCy-MSC stem cells, with the latter consequently being more oriented toward osteogenesis than DPSD, which would seem to be more oriented toward dentinogenesis [240]. Despite being considered biological waste products, hPCy-MSC have a potent capacity for differentiation toward neurogenesis and osteogenesis. In fact, the exosomes derived from hPCy-MSC freed in a dopaminergic neuronal culture medium may offer improvements for early diagnosis as well as new therapies for Parkinson’s disease [238]. One of the principal advantages of this cell population is its easy harvesting and the necessary access to its extirpation without influencing healthy tissues, as well as its alternative potential in regenerative medicine [238].

### 5.13. MSC Derived from Dentigerous Cysts (DCMSC)

The dentigerous cyst (DC) is a disease that destroys the bone as a consequence of the accumulation of fluid between the crown of the unerupted tooth and the reduced enamel epithelium in the embryonic stage (Figure 2). This is due to the fact that the capsule adheres to the cemento-enamel junction that is visualized by X-rays, whose treatment lies in the surgical enucleation of the cyst (including its capsule and the affected tooth), which usually induces dentofacial deformity, including pathological fracture. It represents approximately 20% of odontogenic cysts among the most common cysts in the oral and maxillofacial regions [241]. Previously, as described in the previous section, Marelli et al. isolated MSC-like cells from human periapical cysts [59,237]. Therefore, these previous findings supported the possibility of the presence of stem cells in the DC capsule, assigned the acronym DCCMC. In fact, Yu et al. recently isolated two types of MSCs from the DC fibrous capsules of five patients, both before (Bm-DCSC) and after (Am-DCSC) marsupialization [242]. The authors reported positive expression for CD44 and CD90 cell surface antigens and negative expression for hematopoietic antigens (CD34– and CD45–). MSC-like Am-DCSCs expressed STRO-1 adjacent to bone tissue. In order to corroborate the ability to repair the bone defect resulting from DC, subcutaneous ectopic osteogenesis was tested on a model of cranial bone defect in mice. The results determined that cells with MSC characteristics could be isolated from DC capsules, both before and after marsupialization. However, Am-DCSCs demonstrated a better capacity for proliferation and self-renewal than Bm-DCSCs. Even better capacity to repair the bone defect was confirmed with greater capacity for ectopic bone regeneration. Following the results of this clinical trial, it stands out that MSCs reside in the DC capsule, and marsupialization improves osteogenic capacity both in vivo and in vitro [242]. This discovery provides a scientific basis for the treatment of DC after marsupialization and enables seamless subsequent orthodontic treatment as well as implant placement, paving the way for a multidisciplinary dental approach.

### 5.14. Mesenchymal Stem Cells from Bichat’s Fat Pad (BFPSC)

Mesenchymal cells derived from the fat in the sacs or adipose sacs of Bichat, BFPSC (Bichat’s fat pad stem cell), were isolated and quantified in the perioral fat by Farré-Guasch et al. [58]. They observed that the specialized Bichat fatty tissue had a different origin than subcutaneous fat and was abundantly irrigated, making it an ideal resource given its easy and accessible harvesting via a small incision under local anesthesia with minimal morbidity for the donor. In their study, they found that the Adipose tissue stem cells (ADSC) in the BFP expressed the characteristic ADSC markers (CD 29, CD34, CD73, and CD90) and could differentiate into chondrocytes, osteoblasts, and adipocytes. However, the BFPSC were slightly better (30%) than the ADSC from subcutaneous adipose tissue (22%) [58]. ADSC are MSC that have some 100 to 500 times more SC than bone marrow and can be injected, encapsulated in biomaterials, or implanted in wounds since they improve healing and can directly differentiate into specific cell lines [243]. Therefore, although BM-MSC have been successfully employed in trauma and general surgery (tumors, malformations, etc.), ADSC present more advantages given the amount of tissue present and the less invasive procedure for their harvesting and isolation. At the same time, CD34 expression is characteristic of fresh ADSC, and this expression decreases with the cycles in contrast to MSC from bone marrow that lack this marker and can stimulate angiogenesis, promoting healing in ischemic or graft tissues in oral surgery. In light of this, BFPSC are considered an excellent source of fresh ADSC that avoids the need for in vitro expansion, the high cost, and the contamination risk of clinical application for tissue engineering to repair cartilage and bone defects [58]. What is more, the size of BFP as perioral tissue seems to be similar in different individuals and independent of their body weight and fat distribution, making it even more appropriate as a predictor of insulin resistance. On the contrary, cell diameter seems to influence adipocyte dedifferentiation efficiency, with the smallest (40 μm) showing greater efficiency and consequently greater osteogenic potential than adipocytes with a greater diameter (100 μm) [244]. In a recent systematic review, Gaur et al. concluded that ADSCs can be specifically applied for bone tissue engineering in the treatment of alveolar bone defects in dental implants and periodontal disease due to their potential to differentiate into the periodontal ligament, cementum, and pulp tissue [245]. Nevertheless, MSCs, with a greater differentiation capacity and less risk, are preferred. In fact, an experimental study compared the chondrogenic and osteogenic differentiation capacity of BFPSC to that of cells derived from gum or gingival tissue (GMSC) and had similar results, although the BFPSC are preferred given their lower risk of contamination from the normal oral flora and their lower level of fibroblasts when compared with GMSC [246]. From the osteoblastic differentiation perspective, the fat cells de-differentiated from BFP showed a greater capacity than those of the SC from the same BFPSC [247]. It should be noted that ADSC are a heterogeneous population, particularly during the first culture passes, which can easily be found in the stromal vascular fractions (SVF) due to the disassociation of the adipose tissue. On the other hand, the possibility of BFPSC differentiating under coculture with fibroblasts derived from human salivary glands led to the successful differentiation of these cells later. In this way, Kawakami et al., after managing to obtain salivary gland cells, undertook to transplant them and achieve new tissues so as to advance in improving the quality of life in those patients who have salivary gland atrophy or hypofunction [248]. Also, another study started with BFPS and the differentiation into induced neural cells that were later transplanted in a rat hemiparkinsonian model and found that the cells survived as dopaminergic neural cells and no tumors were generated [249]. The authors suggest that the neurons differentiated from BFPSC could be applied to cell replacement therapy in Parkinson’s disease. In general, stem cells from peribuccal fat are easy to obtain and represent an important source of cells that differentiate according to the needs of regenerative medicine and dentistry, although their application requires further confirmation in human clinical trials.

### 5.15. Salivary Gland-Derived MSC (SGSC)

Stem cells from salivary glands, known as SGSC, were isolated from the submandibular rat salivary gland; they were highly proliferative and had the capacity to express acinar, myoepithelial, and ductal cell line markers [250]. However, it is difficult to understand the biological properties of isolated progenitor and stem cells because they are mixed and heterogeneous [251]. This is due to the fact that the salivary gland is composed of two main epithelial compartments: the ducts, which transport and modify saliva, and the acinar cells, which produce saliva. These cells are surrounded by a stromal matrix containing contractile myoepithelial cells, fibroblasts, immune cells, endothelial cells, and neurons. Like the liver, prostate, and lung, the salivary gland belongs to a group of tissues in our body with a relatively slow replacement rate that is below 60 days. These tissues increase their proliferation rate in response to damage so as to replace the lost cells and return to homeostasis [252]. Therefore, the importance of GS-secreted saliva, albeit non-vital, is its ability to maintain physiological equilibrium and initiate alimentary digestion. Flow cytometry analysis revealed positive expression for mesenchymal markers CD29, CD44, CD73, CD90, CD105, and HLA-DR, but negative expression for hematopoietic and endothelial markers [251]. Dysfunctional GS are behind the current interest in the need for new therapeutic focuses that could provide long-term solutions to restore function in different salivary glands. In fact, hyposalivary dry mouth often induces irreversible xerostomia, with symptoms like oral dryness, loss of buccal hygiene, loss of taste, difficulty speaking, chewing, swallowing, etc., which, together, reduce quality of life. GS hypofunction, particularly in the elderly, is part of aging but is made more severe mainly as a result of the secondary effects of drugs like anticholinergics, antidepressants, blood pressure medications, and diuretics that are used to treat systemic diseases. On the other hand, hyposalivation may also be a consequence of autoimmune disorders like Sjögren’s disease, endocrine disorders (diabetes mellitus and thyroid disease), neurological alterations, or radiation damage in head and neck cancer patients after radiotherapy [21]. Generally, treatment options for xerostomia include the administration of saliva substitutes or stimulants (if there is any GS function). Also, the hyposalivation frequently experienced by people receiving radiotherapy for head and neck cancers has a large number of causes including the lack of functional acinar cells to produce saliva, a result of radiation-induced stem cell sterilization [253]. Human SGSC exhibit epithelial and mesenchymal phenotypes as well as a multipotent differentiation potential, which could regenerate radiation-damaged SG [251]. Stem cells with a mesenchymal-like phenotype have been isolated from human salivary glands [21]. Producing functional salivary glands from rat embryonary stem cells is a major challenge. Salivary gland SC transplantation has restored gland function in irradiated rats, suggesting that a therapy with SC could be helpful against the consequences of dry mouth in patients. In fact, Nanduri et al. demonstrated that SGSC isolated in rats were able to form structures like the neurospheres called sialiospheres. Consequently, SGSC could be isolated from biopsies and expanded in vitro, increasing their number to form sialiospheres [254]. Human sialospheres have been found in human salivary gland (SC) cultures. Additionally, these cells are able to renew themselves with the capacity to differentiate on transplantation into irradiated rats, demonstrating in vivo that the cell transplant could revert gland function. Thus, based on this study by Pringle and collaborators, it can be affirmed that suprarenal marrow SC cultured from biopsies can revert hyposalivation induced by radiation itself [253]. A revision study describes the use of genetic line tracing in rats since cells can be identified and isolated based on their protein and enzyme expression in different circumstances (in vitro floating sphere assays, 2D and 3D cultures in humans). These progenitor and stem cells were present in the different developmental stages of the organ and could compensate for cell loss, enabling the appropriate formation of the organ. Even during homeostasis of salivary glands in adults, the multiple cell-type reservoirs in the compartments have the capacity to duplicate, maintain, or expand their number [255] of epithelial cells in the ectoderm-derived salivary gland [256]. In fact, there is no clear consensus on the criteria to be applied to identify these supposed SGSC. Even so, therapy with SC replacement can be a good option for treating radiation-induced hyposalivation. Identification of stem and progenitor cell populations in the salivary gland allows the use of stem cells to treat xerostomia and prevent the longer-lasting form of hyposalivation [255]. At any rate, the capacity to proliferate or differentiate in vitro and form spheres could rescue salivary gland function after this transplant of irradiated glands. On the contrary, the results vary in the different assays and suggest caution in comparing the published results. Up to the present, no cell that fulfills the strict definition of a stem cell has been certainly/undoubtedly/indisputably identified in adult salivary glands. Therefore, there are several cell populations in adult glands that have different potentialities to proliferate, differentiate, and have a differential capacity to be stimulated by soluble paracrine factors [245]. Perhaps the therapeutic approaches should consider the specific identity of a stem cell and focus more on a cell state that can be manipulated. Lastly, a critical area for investigation is delving into understanding the cell-cell interactions involved in plasticity (particularly in questions referring to salivary glands) [257].

### 5.16. Tonsil-Derived Mesenchymal Stem Cells (TMSC)

The stem cells of the palatine tonsils, designated by the acronym TMSC (Human Tonsil-derived Mesenchymal Stem Cells), due to their promising advantages, were isolated and cultured in different states (fresh, cryopreserved, and thawed). They demonstrated a high proliferative capacity and expressed primitive markers of the cell surface of the MSCs, CD73, CD95, and CD105, in the absence of hematopoietic markers; they even presented their own distinctive markers, such as CD106 and CD166, which are closely related to adhesion, migration, and immunomodulation [258,259]. Human tonsils are found in the oropharynx and are composed of lymphoepithelial tissue with immunological activity that decreases after puberty [258]. They are considered waste tissue; therefore, they lack the limitations present in the necessary invasive procedures for obtaining SCs from bone marrow (gold standard in MSC research applications), adipose tissue, and cord umbilical blood, among others. Furthermore, these sources are mostly of mesodermal origin and are likely to present obstacles to differentiation in ectodermal and endodermal tissues [258]. In this sense, TMSCs show a relatively higher yield and proliferation, with differentiation potential, not only in mesodermal lineages (bone, muscle, cartilage, tendon, and fat), but also in endodermal lineages (hepatocytes, secreting cells of insulin, and parathyroid cells) and ectodermal lineages (glial and neuron-like cells). Likewise, they stand out for their viability after cryopreservation, since the ideal is for the patient to receive a transplant of their own MSCs, which are not always available when the need arises [176,258,259]. It should be noted that most tonsillectomies are performed between the ages of 5 and 19, which means that TMSCs come from young donors. In this sense, TMSCs are characterized by a higher proliferation rate, with an average doubling time of 37 h compared to 58 h for BM-MSC. Also, they are quite safe and stable up to pass 15 for in vitro expansion, compared to BM-MSC, which are good and reliable below pass five. However, it is quite safe and stable up to passage 10 for application in future clinical settings [260]. Lee Hyun-Ji conducted a study in mouse models subjected to a myelosuppressive regimen in order to test strategies to improve the results of bone marrow transplantation. The most widely used therapy for patients with hematological malignancies is hematopoietic SC transplantation, despite presenting a mortality rate of 50 to 60%. In fact, MSC co-transplant is another alternative with better results. Therefore, the researchers used TMSC as a co-transplant and showed that TMSC improved mouse survival, bone marrow engraftment, and blood cell recovery; in contrast, these effects were not observed when MMP3 (metalloproteinase) expression was downregulated in TMSCs. Taken together, these results suggest that MMP3 expressed in TMSC enhances bone marrow engraftment by degrading type IV collagen in the extracellular matrix, which facilitates its migration and reduces the duration of pancytopenia [261]. TMSCs have recently been successfully introduced as a possible therapeutic alternative for various diseases, including inflammatory bowel diseases, due to their immunosuppressive and tissue-regenerative properties. The results on the murine model of acute and chronic colitis indicated that TMSCs cultured in 3D showed considerably higher therapeutic effects than those cultured in 2D through increased expression of anti-inflammatory cytokines, leading to improvement in clinical symptoms and loss of body weight, among others [262]. Likewise, the results were encouraging in terms of the efficacy of wound healing. A recent study concluded that the regenerative potential of TMSCs was produced through immunomodulation and regeneration of the dermis and epidermis in the murine excision splint model. Similarly, data on atopic dermatitis in the mouse model were published, showing particularly therapeutic rather than protective effects. Based on in vivo and in vitro results, the therapeutic potency of TMSCs lay in the inhibitory effects of T and B cell-mediated inflammatory responses, decreased levels of IL-6, IL-1β, TNF-α, IL-4, and serum IgE, whereby inflammatory skin lesions were improved by subcutaneously injected TMSCs [263]. Currently, with respect to TMSC, no clinical trials have been initiated, but some preclinical trials are being carried out to try to recover neural, muscular, and parathyroid functions. In this sense, the evidence postulates that the phenotypic and functional characteristics of TMSCs suggest that they could potentially be designated as a great source of MSCs for future stem cell engineering in regenerative medicine, in consideration of stable clinical practice.

## 6. Challenges and Opportunities for Future Therapeutic Applications: Future Directions

According to the reported results in the clinical trials with dental mesenchymal stem cells (DMS), there is great potential for clinical treatments. However, the small number of enrolled patients in some clinical trials is not concordant with the growing scientific basis of research in the regenerative medicine field. This clearly translates into the need for more efforts and multidisciplinary coordination among basic and clinical researchers in order to make possible the MSCs application in patients. The benefits obtained from biotechnological advances to treat endodontics, oral surgery, or periodontics allow the use of less invasive treatments in dentistry. In fact, in the quest to accelerate the translation of discoveries from the bench to the clinic and communities, research centers for clinical and translational sciences have been developed. This aim pursues the promotion and development of solid growth both in studies and in laboratory infrastructures to streamline clinical work and finally reach therapeutic practice, improving community well-being and reducing health disparities [264,265,266,267]. The implementation of tooth banks, until now absent in Spain, could guarantee the cryopreservation of oral cells and tissues as an option from an early age, both for autologous and allogeneic use, even with the use of culture supernatants and released exosomes by the MSCs. Faced with stem cells from superior lineages, the scientific community continually strives to obtain and maximize the use of adult stem cells in new disease models and discover early markers or novel therapeutic approaches, improving the diagnosis and prognosis of various human pathologies free of bioethical and moral conflicts (Figure 3).

## 7. Conclusions

Among the general strengths and weaknesses of studies of adult stem cells of dental origin and surrounding areas, the scientific community continues its efforts to understand the pathways and biology of molecular processes and the interaction with biomaterials in the engineering of tissues and lines that lead to healing. Both in medicine and regenerative dentistry, progress is slow and expensive as a result of the safety that is implemented in each process and evaluation on the same basis of action towards clinical translation, trusting that in the future there will be less complex and expensive treatments but more unified, accessible, and applicable in the different stages of a person’s life. It is necessary to encourage the promotion and support of research on the use of adult stem cells since they do not imply ethical and moral problems [80].

The consequences of the culture surrounding the push for publication increase the acknowledgment of current research, but sometimes its clinical translation is questionable. The publications in high-impact journals and the use of citation metrics could help researchers and clinics develop a better practice in their clinics [267] for those who find themselves juggling responsibilities outside of the research domain; thus, stem cell treatments for patients must be safe; additionally, the current academic reward system and institutional structures that are in place to reward translational researchers should be improved in our country (Spain). Ultimately, we hope that this review will contribute to increasing the recognition of oral cavity stem cell therapy and its clinical application. Finally, we emphasize that the biobank enhances the future biomedical application of stem cells and also increases collaborations between cell therapy in basic dentistry and its clinical applications. It is necessary to encourage the promotion and support of the field of adult stem cell research, free of ethical and moral conflicts [80].

## Figures and Tables

**Figure 1 pharmaceutics-15-02109-f001:**
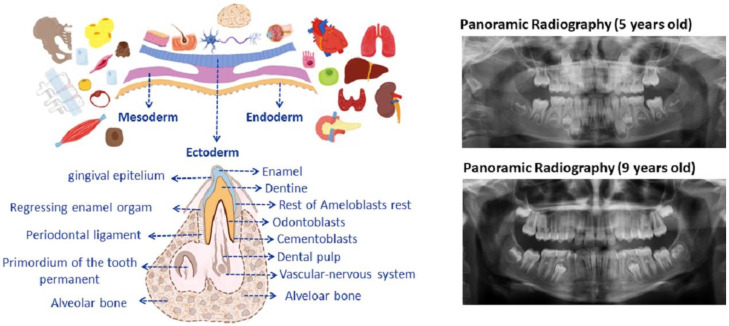
Development of germinal layers: ectoderm, mesoderm, and endoderm (**left**) and panoramic radiography of children (5 and 9 years old, (**right**)). A deciduous incisor in partial eruption next to the primordium of the permanent incisor, which will subsequently replace it, and parts of teeth and panoramic radiography of children (5 and 9 years old). Adapted from Yamada et al. [27].

**Figure 2 pharmaceutics-15-02109-f002:**
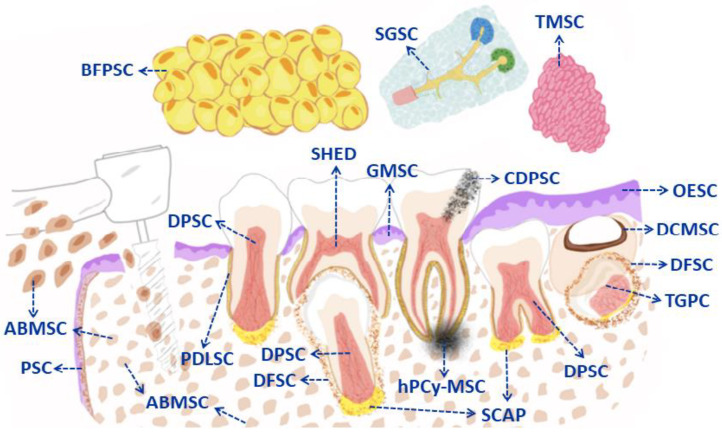
Kinds of mesenchymal stem cells in the oral cavity. **ABMSC**: alveolar bone mesenchymal stem cell; **DPSC**: dental pulp stem cell; **SHED**: stem cells from exfoliated deciduous teeth; **SCAP**: apical papilla-derived MSC; **CDPSC**: carious dental pulp stem cells; **PSC**: periosteum-derived MSC; **SGSC**: salivary gland-derived MSC; **TMSC**: MSC derived from the palatine tonsils; **PDLSC**: stem cells derived from periodontal dental ligament; **hPCy-MSC**: MSC derived from periapical cysts; **OESC**: oral epithelium derived MSC; **DCMSC**: MSC derived from Dentigerous Cysts; **DFSC**: dental follicle-derived MSC; **TGPC:** tooth germ progenitor-derived MSC; **PDLPs**: periodontal ligament progenitor cells; **GMSC**: gingival stem cells; **BFPSC**: Bichat’s fat pad derived.

**Figure 3 pharmaceutics-15-02109-f003:**
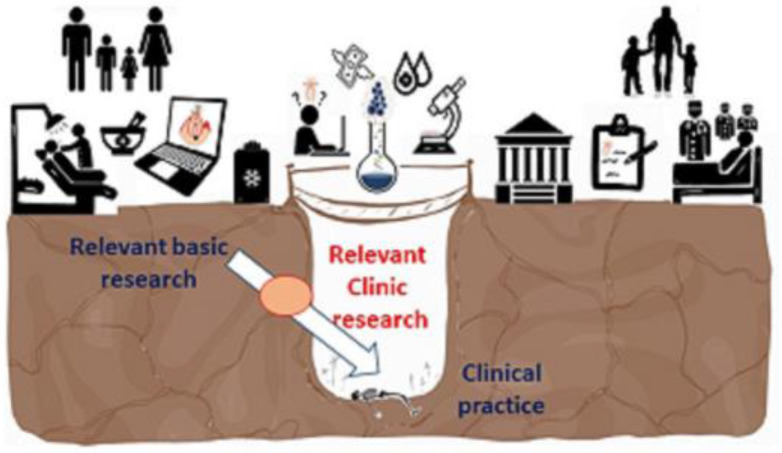
Bridge between basic and clinical biomedical applications. Imbalance in the generation of knowledge from basic research to clinic practice. Adapted from Butler D, [265]. Representation adapted from obtaining adult mesenchymal stem cells from the oral cavity and surrounding areas, their cryopreservation, research, and development; the safety and efficacy of clinical procedures designed for patients; legislation; and translational application. An attempt to bring basic research closer to clinical application.

## Data Availability

Not applicable.

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
