# Peer review of "Adult Mesenchymal Stem Cells from Oral Cavity and Surrounding Areas: Types and Biomedical Applications"

_pharmaceutics, 2023, doi:10.3390/pharmaceutics15082109_

Round 1
Reviewer 1 Report
This is a very lengthy review of the different types of mesenchymal stem cells that can be found in the oral cavity, pages 1-19, and summarised in Table 2. This is followed by a section on the translational use of these cells, p20-25, with Tables 3 & 4 describing various clinical trials with these cells. Their use in a variety of different medical conditions unrelated to oral health, e.g., ocular pathology, liver disease, diabetes, cardiovascular disease, neurological diseases, p 31-35. Biobanking and cryopreservation of the cells can be found on pages 35-42.
The content is far too long for a standard review. Indeed, one wonders if the material has been written for another purpose, such as a long book chapter, and the authors are trying to maximise its usefulness. The authors should clarify this point since if it has been published or is being published elsewhere, it should not also be published in a peer reviewed journal.
If this is pristine material, it needs to be shortened considerably. The section on Biobanking should be deleted – it could be the subject of a separate submission. A description of the different types of stem cells is fine, but rather than having different sections thereafter on clinical trials and different medical conditions, the information in these latter sections might be included with the specific types of stem cells themselves. This would shorten the publication and reduce the inordinately long list of references.
The information in Tables 3 and 4 re clinical trials is mostly taken from reference 236. Is there really a need to reproduce it since it has already been published?
Minor changes:
The abbreviation PDSC is used on a number of occasions, but this might be a mistype for DPSC, dental pulp stem cells. This should be corrected.
A glossary of abbreviations should be included.
Author Response
Comments to the reviewer-1
Dear reviewer
Thanks for your comments, which improves the quality of this R1 version
This is a very lengthy review of the different types of mesenchymal stem cells that can be found in the oral cavity, pages 1-19, and summarised in Table 2. This is followed by a section on the translational use of these cells, p20-25, with Tables 3 & 4 describing various clinical trials with these cells. Their use in a variety of different medical conditions unrelated to oral health, e.g., ocular pathology, liver disease, diabetes, cardiovascular disease, neurological diseases, p 31-35. Biobanking and cryopreservation of the cells can be found on pages 35-42.
The content is far too long for a standard review.
We have shortened the content in this R1 version following your advice. Thanks again¡
Indeed, one wonders if the material has been written for another purpose, such as a long book chapter, and the authors are trying to maximise its usefulness. The authors should clarify this point since if it has been published or is being published elsewhere, it should not also be published in a peer reviewed journal.
This R1 content is not published elsewere and is not under consideration in another journal. We confunded the word chapter by review. Sorry¡.
If this is pristine material, it needs to be shortened considerably. The section on Biobanking should be deleted – it could be the subject of a separate submission.
This is a good suggestion. We have remove it the biobank section.
A description of the different types of stem cells is fine, but rather than having different sections thereafter on clinical trials and different medical conditions, the information in these latter sections might be included with the specific types of stem cells themselves. This would shorten the publication and reduce the inordinately long list of references.
Done it¡. We have included description of each MSC stem cells type from the oral cavity, incuding its biomedical applications.
The information in Tables 3 and 4 re clinical trials is mostly taken from reference 236. Is there really a need to reproduce it since it has already been published?
We agree with you. Therefore, these tables 2, 3 and 4 have been removed in this R1 version.
Minor changes:
The abbreviation PDSC is used on a number of occasions, but this might be a mistype for DPSC, dental pulp stem cells. This should be corrected.
Yes, PDSC was replaced by DPSC in the R1 version
A glossary of abbreviations should be included.
Done it¡. We included a glossary at the end of discusión following your advice.

Reviewer 2 Report
Dear Authors.
The paper submitted for review is, in my opinion, the most thorough and lengthy review of oral stem cell research. After reading it, I have only 3 comments:
1. the paper should be formatted according to the requirements of the elective
2. the chapter on tooth development should be shortened,
3. improve the quality of the figures
4. please consider whether the descriptions of some of the cell types that can be isolated from the oral cavity but are not relevant to regenerative medicine should be significantly shortened, the paper should be shorter maximum 35 pages
Author Response
Reviewer2-response
Dear reviewer-2.
The manuscript content has been improved it follwing your advice. Thanks a lot¡
The paper submitted for review is, in my opinion, the most thorough and lengthy review of oral stem cell research. After reading it, I have only 3 comments:
Thanks for your comments, which improve the quality of this R1 version
- the paper should be formatted according to the requirements of the elective
Done it¡. Thanks for your advice. The abstract has 200 works.
- the chapter on tooth development should be shortened,
We have shortened the content of this R1 version. In addiiton, tables 1, 2 and 3 are not included because reviewer-1 suggest its removal. This is his/her textually commentary ¨The information in Tables 3 and 4 re clinical trials is mostly taken from reference 236. Is there really a need to reproduce it since it has already been published?¨. This is reason by which these tables are not included in this R1 version. In addition, the biobank section was removed following reviewer-1 advice. This is his/her textual comment ¨ The section on Biobanking should be deleted – it could be the subject of a separate submission¨.
- improve the quality of the figures
Figures are pasted as TIFF format in this R1 version. The quality was improved now at 300 dpi (figure -1=832 Kb, figure 2=1437 Kb, figure 3=1993 Kb).However, you must take into account that figures must be included in the Word within the MDPI submission system; as as consequence, its quality could be reduce it. We also sent figures in TIFF format to the Journal in order to avoid problems of resolution.
Dear reviwer-2, thanks again for your comment.
- please consider whether the descriptions of some of the cell types that can be isolated from the oral cavity but are not relevant to regenerative medicine should be significantly shortened, the paper should be shorter maximum 35 pages
Done it¡. We have done efforts and the manuscript has been sorthened in this R1 version.
